# Synaptic pruning in the female hippocampus is triggered at puberty by extrasynaptic GABA$_A$ receptors on dendritic spines

Sonia Afroz[1,2†‡], Julie Parato[1,2†], Hui Shen[1,3], Sheryl Sue Smith[1,4*]

[1]Department of Physiology and Pharmacology, SUNY Downstate Medical Center, Brooklyn, United States; [2]Program in Neural and Behavioral Science, SUNY Downstate Medical Center, Brooklyn, United States; [3]School of Biomedical Engineering, Tianjin Medical University, Tianjin, China; [4]The Robert F. Furchgott Center for Neural and Behavioral Science, SUNY Downstate Medical Center, Brooklyn, United States

*For correspondence: sheryl.smith@downstate.edu

[†]These authors contributed equally to this work

Present address: [‡]Department of Biomedical Sciences, University of California, Riverside, United States

Competing interests: The authors declare that no competing interests exist.

**Abstract** Adolescent synaptic pruning is thought to enable optimal cognition because it is disrupted in certain neuropathologies, yet the initiator of this process is unknown. One factor not yet considered is the α4βδ GABA$_A$ receptor (GABAR), an extrasynaptic inhibitory receptor which first emerges on dendritic spines at puberty in female mice. Here we show that α4βδ GABARs trigger adolescent pruning. Spine density of CA1 hippocampal pyramidal cells decreased by half post-pubertally in female wild-type but not α4 KO mice. This effect was associated with decreased expression of kalirin-7 (Kal7), a spine protein which controls actin cytoskeleton remodeling. Kal7 decreased at puberty as a result of reduced NMDAR activation due to α4βδ-mediated inhibition. In the absence of this inhibition, Kal7 expression was unchanged at puberty. In the unpruned condition, spatial re-learning was impaired. These data suggest that pubertal pruning requires α4βδ GABARs. In their absence, pruning is prevented and cognition is not optimal.

## Introduction

During the pubertal period, the density of dendritic spines decreases by half in widespread areas of the CNS (*Huttenlocher, 1979*; *Zehr et al., 2006*; *Petanjek et al., 2011*; *Koss et al., 2014*), including the CA1 hippocampus and temporal lobe of both rodents (*Yildirim et al., 2008*) and humans (*Tang et al., 2014*), sites essential for learning and memory (*Pastalkova, 2006*). Dendritic spines express NMDA receptors (NMDARs) at excitatory synapses (*Matsuzaki et al., 2004*) which can be activated to form memory traces (*Bannerman et al., 2008*). A modelling study (*Ruppin, 1999*) suggests that an optimal spine density, produced by developmental pruning of unnecessary synapses, may be necessary not only for the ability to form memories but also the ability to re-learn or 'update' previously learned information. Despite the implied importance of synaptic pruning during adolescence, the mechanisms underlying spine elimination in CA1 hippocampus during puberty are not yet known nor are the behavioral outcomes of altered spine density.

At certain times in development, scavenging by immune system components such as the microglia plays a role in spine pruning (*Schafer et al., 2012*; *Sekar et al., 2016*). This system is likely the final step in synapse elimination in several CNS areas including the CA3 hippocampus, but does not have a role in synapse loss of CA1 hippocampal pyramidal cells during puberty (*Shi et al., 2015*).

**eLife digest** Memories are formed at structures in the brain known as dendritic spines. These structures receive connections from other brain cells through regions called synapses. In humans, the number of these brain connections increases dramatically from birth to childhood, reflecting a period of rapid learning. However, the number of these brain connections halves after puberty, a dramatic reduction shown in many brain areas and for many species, including humans and rodents. This process is referred to as adolescent synaptic pruning and is thought to be important for optimal learning in adulthood because it is disrupted in autism and schizophrenia. Synaptic pruning is believed to remove unnecessary brain connections to make room for new relevant memories. However, the process that triggers synaptic pruning is not known.

Within the brain, proteins called inhibitory GABA receptors are targets for chemicals that reduce the activity of nerve cells. As brain connections must be kept active to survive, inhibitory receptors could help to trigger synaptic pruning.

Afroz, Parato et al. now show that, at puberty, the number of a particular type of $GABA_A$ receptor increases in the brain of female mice. This triggers synaptic pruning in the hippocampus, a key brain area necessary for learning and memory. By reducing brain activity, these inhibitory receptors also reduce the levels of a protein in the dendritic spine that stabilizes the scaffolding of the spine to maintain its structure.

Mice that do not have these $GABA_A$ receptors maintain a constant high level of brain connections throughout adolescence, and synaptic pruning does not occur in their brains. These mice were initially able to learn to avoid a specific location that provided a mild shock to their foot. However, when this location changed the mice were unable to re-learn where to avoid, suggesting that too many brain connections limits learning potential.

Brain connections are regulated by many factors, including the environment and stress. Future studies will test how these additional factors alter synaptic pruning in adolescence, and will test drugs that target these inhibitory receptors to manipulate adolescent pruning. These findings may suggest new treatments for "normalizing" synaptic pruning in conditions where this process occurs abnormally, such as autism and schizophrenia.

One factor not yet considered in adolescent pruning is the role of inhibition in the brain mediated by $GABA_A$ receptors (GABARs). GABARs mediate most inhibition in the brain and are pentameric membrane proteins, of diverse sub-type, which conduct a $Cl^-$ current. In the hippocampus, GABARs are either expressed sub-synaptically, where they generate a phasic current, or extrasynaptically, where they underlie a tonic inhibitory conductance (*Stell and Mody, 2002*).

During the pubertal period (PND 35–44) of female mice, we have shown that $\alpha4\beta\delta$ GABARs transiently emerge on dendritic spines of CA1 pyramidal cells, adjacent to excitatory synapses (*Shen et al., 2010*; *Aoki et al., 2012*). These extrasynaptic receptors, which are sensitive to low levels of ambient GABA (<1 μM) (*Brown et al., 2002*), generate a shunting inhibition which reduces NMDAR-generated current (*Shen et al., 2010*). However, NMDA current is robust in pubertal mice lacking expression of $\alpha4\beta\delta$ GABARs (*Shen et al., 2010*), suggesting that the inhibition mediated by $\alpha4\beta\delta$ GABARs produces the impairment rather than a lack of functional NMDARs at puberty.

We have also shown that the pubertal increase in hippocampal $\alpha4\beta\delta$ GABAR expression prevents induction of long-term potentiation, an in vitro model of learning, and impairs spatial learning of female mice (*Shen et al., 2010*; *Shen et al., 2016*). These deficits were not observed at puberty in α4 KO (*Shen et al., 2016*) or δ KO (*Shen et al., 2010*) mice, implicating pubertal $\alpha4\beta\delta$ GABARs as the mediating factor.

We have extended these studies to show here that expression of $\alpha4\beta\delta$ GABARs at the onset of puberty initiates synaptic pruning in the female mouse hippocampus, which ultimately reduces spine density post-pubertally (comparing spine density at PND 35 versus PND 56). In the α4 KO mouse, pruning does not take place and the cognitive ability of the post-pubertal mice is impaired. We also show $\alpha4\beta\delta$ and NMDAR involvement in the pruning process by the administration of selective drugs during the pubertal period (PND 35–44) to determine effects on spine density post-pubertally (PND

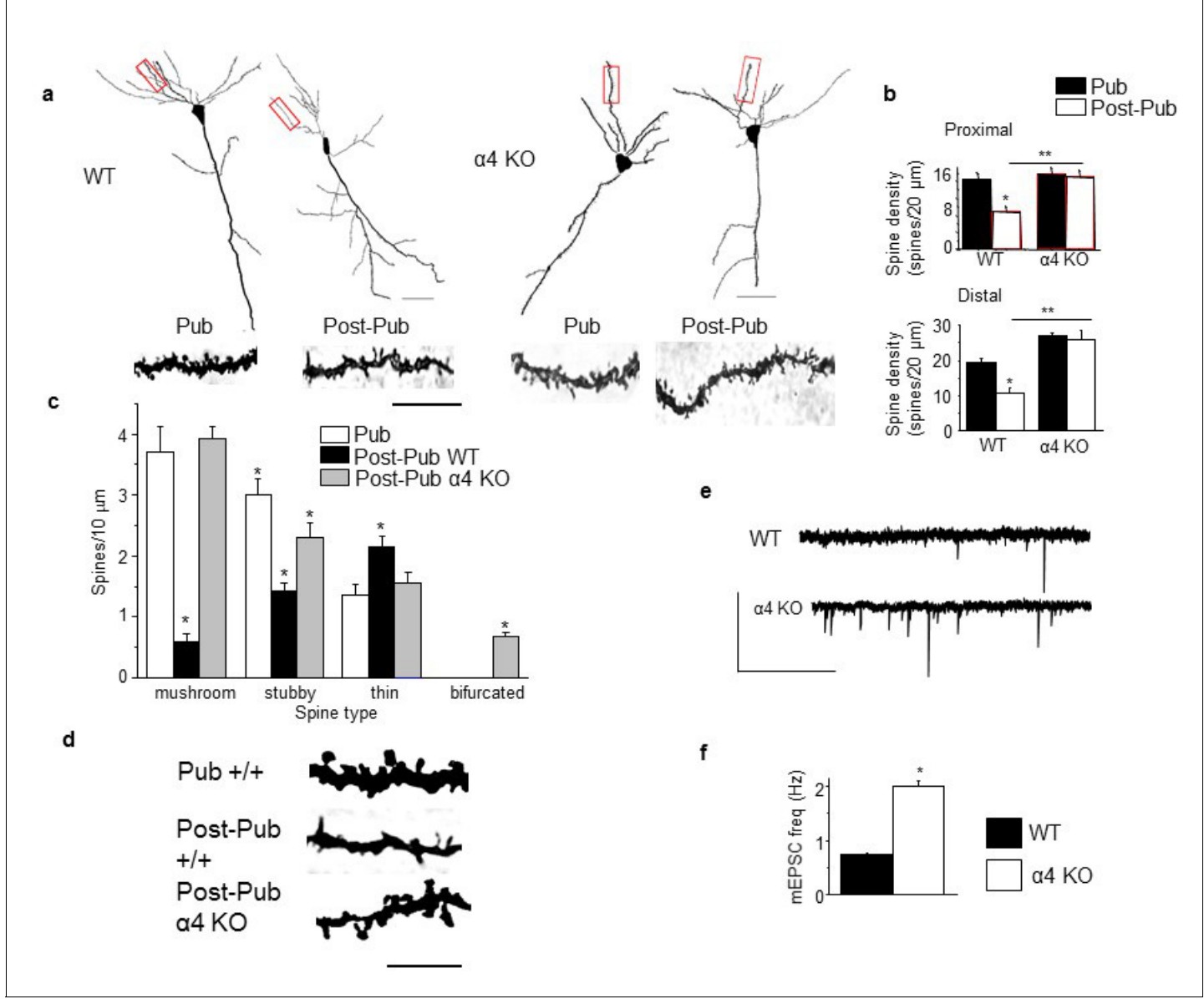

**Figure 1.** Synaptic pruning of CA1 hippocampus of adolescent female mice is prevented in the α4 knock-out. Pub, pubertal; post-Pub, post-pubertal. (a) CA1 hippocampal pyramidal cells, Pub and post-Pub (8-week old) WT and α4 KO female mouse hippocampus. Upper panel, neurolucida images, scale, 50 μm; lower panel, z-stack (100x) images; scale, 20 μm. Additional images and data from male mice provided in *Figure 1—figure supplement 1* and *Figure 1—figure supplement 2*, respectively. Source data for all figures are available as separate files. (b) Averaged data for spine density, Proximal (left), WT, t-test, t(41)=7.15, p<0.0001*, power=1; n= 21–22 neurons (5–6 mice)/group; α4 KO, t-test, t(47)=0.43, P=0.67; n= 24–25 neurons (6 mice)/group; post-Pub, WT vs. α4 KO, t-test, t(45)=5.8, p<0.0001*; Distal (right), WT, t-test, t(28)=5.73, p<0.0001, power=1*; n= 15 neurons (5–6 mice)/ group; α4 KO, t-test, t(39)=2.11, P=0.04; n= 20–21 neurons (6 mice)/group; post-Pub, WT vs. α4 KO, t-test, t(33)=8.1, p<0.0001*. *p<0.05 vs. Pub; **p<0.05 vs. WT. (*Figure 1—source data 1*) (c) Quantification of spines according to type, *p<0.05 vs. other pubertal/genotype groups. Mushroom, ANOVA, F(2,54)=110.65, p<0.0001*, power=1; Stubby, ANOVA, F(2,54)=23.1, p<0.0001, power=1; Thin, ANOVA, F(2,54)=9.29, p=0.0003*, power=0.94; Bifurcated, ANOVA, F(2,54)=39, p<0.0001*, power=1; (n=19 neurons, 5 mice/group). *p<0.05 vs. other groups. (*Figure 1—source data 2*) (d) Representative high-contrast z-stack images; scale, 10 μm. (e) Representative mEPSCs, post-Pub WT and α4 KO. Scale, 50pA, 10 s. (f) Averaged data, mEPSC frequency; *t-test, t(16)=11.4, p<0.0001*, power=1; n= 8–10 cells (mice)/group. (*Figure 1—source data 2*)

The following source data and figure supplements are available for figure 1:

**Source data 1.** Spine counts/20 μm on dendrites of CA1 hippocampal pyramidal cells for *Figure 1b* for wild-type (WT) and α4 knock-out (KO) female mice assessed at puberty (Pub, PND 35, identified by vaginal opening) and post-puberty (Post-pub, PND 56).

*Figure 1 continued on next page*

*Figure 1 continued*

**Source data 2.** Spine counts/10 µm for different spine-types on dendrites of CA1 hippocampal pyramidal cells for *Figure 1c* for Pub and Post-pub WT and Post-pub α4 KO.
**Source data 3.** Figure 1f. mEPSC frequency, # mEPSCs/s recorded from CA1 hippocampal pyramidal cells using whole cell patch clamp techniques for post-pubertal WT (left) and α4 KO mice.
**Figure supplement 1.** Neurolucida images of spine density across pubertal stage and α4 genotype.
**Figure supplement 2.** Synaptic pruning of CA1 hippocampus of adolescent male mice is prevented in the α4 knock-out.
**Figure supplement 2—source data 1.** Spine counts/20 µm on dendrites of CA1 hippocampal pyramidal cells for *Figure 1b* for wild-type (WT) and α4 knock-out (KO) male mice assessed at puberty (Pub, PND 35) and post-puberty (Post-pub, PND 56).

56). We further suggest that α4βδ-triggered pruning is due to impairment of NMDAR activation which regulates expression of kalirin-7 (Kal7), a Rho guanine nucleotide exchange factor (GEF) important for stabilizing the actin cytoskeleton (*Penzes et al., 2001*).

## Results

### Spine density changes at puberty

Spine density of both proximal and distal dendrites of CA1 pyramidal cells decreased ∼50% from PND 35 (puberty onset) to PND 56 (post-pubertal, $p < 0.05$) in female mice (*Figure 1a,b*, *Figure 1— figure supplement 1*). To test the role of α4βδ GABARs in spine pruning, we examined spine density across the same age range in α4 KO mice. (Mice with both alleles of the α4 chain of the GABAR gene (*Gabra4*) inactivated are referred to here as α4 KO.) In contrast to the wild-type (WT) mice, there was no significant change in spine density during adolescence in α4 KO mice, for which spine density was 100–150% greater than in WT mice post-pubertally ($p < 0.05$), implicating α4βδ GABARs in spine pruning (*Figure 1a,b*, *Figure 1—figure supplement 1*). Adolescent synaptic pruning was also observed in the male (*Figure 1—figure supplement 2*), where α4βδ GABAR expression is also increased at puberty (unpublished data): Spine density of CA1 hippocampal pyramidal cells decreased by ∼42% from puberty to post-puberty ($p < 0.05$), an effect not observed in the male α4 KO mouse.

Spine morphology was also characterized in the female mouse across pubertal state. Mushroom spines, thought to be 'learning spines' (*Bourne and Harris, 2007*), decreased by ∼85%, while stubby spines decreased by 50% post-pubertally in WT mice ($p < 0.05$); α4 knock-out prevented these changes, resulting in a 500–600% increase in mushroom spines and a 40% decrease in thin spines ($p < 0.05$; *Figure 1c,d*) compared to the WT post-pubertal hippocampus. Dendritic length was unaltered (*Table 1*). Decreased spine density in WT post-pubertal hippocampus was accompanied by decreased frequency of miniature excitatory post-synaptic currents (mEPSCs), reflecting fewer synapses, compared with the α4 KO hippocampus (*Figure 1e,f*).

**Table 1.** Dendrite length is not altered during adolescence or after α4 knock-out. Mean ± SEM, n=4 neurons (mice)/group. Dendrite length, ANOVA, $F_{(3,15)}=0.35$, $p=0.80$.

| Dendrite length (Mean ± SEM) | Pubertal | Post-pubertal |
| --- | --- | --- |
| WT | 190 ± 5.8 | 205 ± 27.5 |
| α4 KO | 183.7 ± 5.5 | 195 ± 9.6 |

Source data 1. Dendrite length for pubertal (Pub) and post-pubertal (Post-pub) WT and α4 KO female mice.

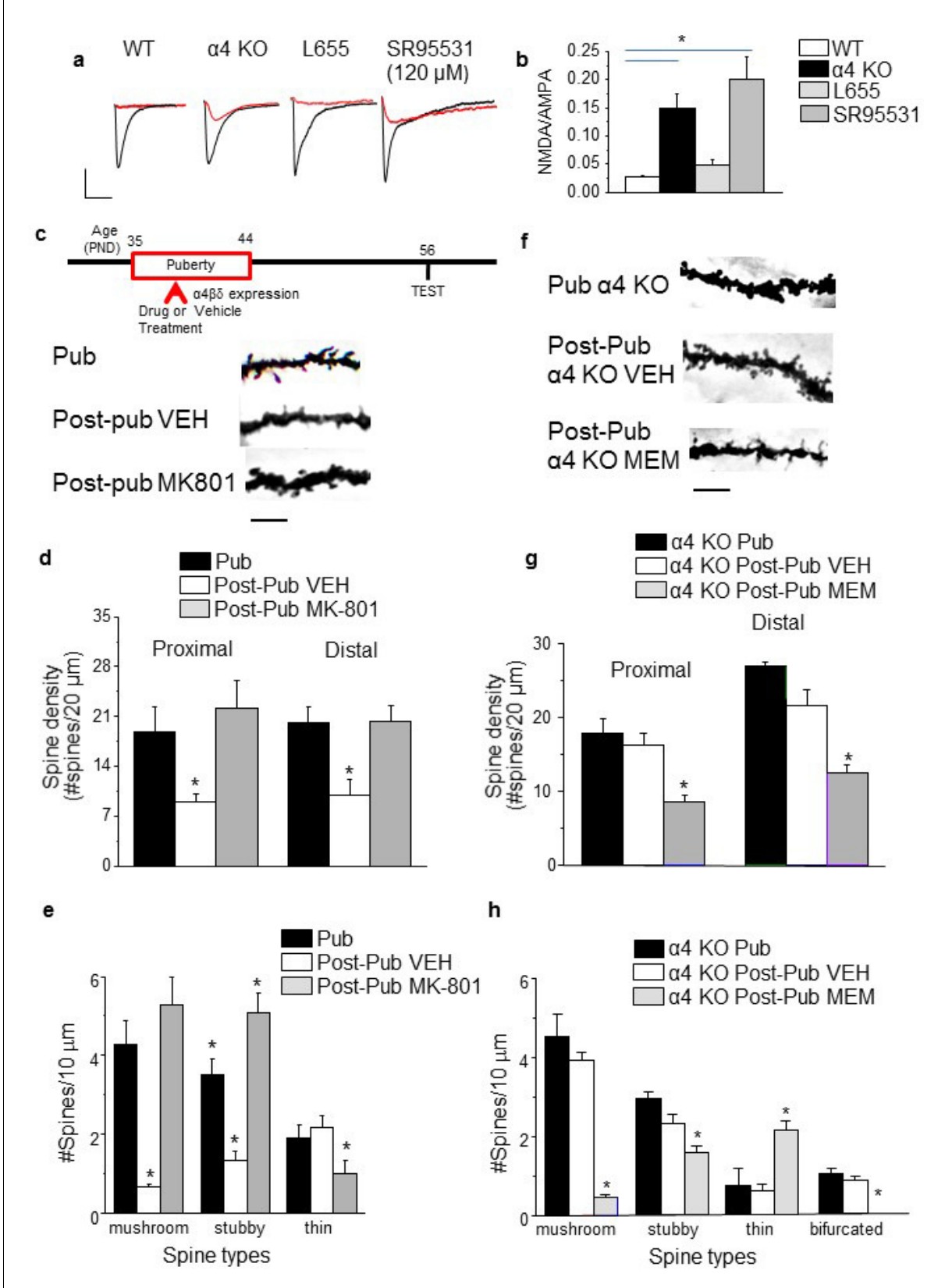

**Figure 2.** NMDA receptors maintain spines during puberty. (a) Representative EPSCs (black) and NMDA EPSCs (red) recorded during puberty in WT or α4 KO hippocampus, in some cases during α5 (50 nM L655) or total (120 µM SR95531) GABAR blockade. In all other cases, 200 nM SR95531 was bath

*Figure 2 continued on next page*

*Figure 2 continued*

applied block synaptic GABARs (*Stell and Mody, 2002*). Scale, 150 pA, 15 ms. (**b**) Averaged NMDA/AMPA ratios; ANOVA, F(3,31)=20.21, p=0.0001*, power=1; n=8–10 cells (mice)/group. (*Figure 2—source data 1*) *p<0.05 vs. WT. (**c**) Inset, Drug treatment during puberty (PND 35–44) was tested for its effect on post-pubertal spine density (PND 56). Z-stack images, pub and post-pub hippocampus, showing the effects of pubertal vehicle or MK-801 treatment, at a dose shown to increase NMDAR expression (*Gao and Tamminga, 1995*). Scale, 6 μm. (**d**) Averaged spine density. Proximal (left): ANOVA, F(2,32)=54.16, p<0.0001*, power=1, n= 11–12 neurons (5 mice)/group; Distal (right)l: ANOVA, F(2,32)=460.1, p<0.0001*, power=1; n=11–12 neurons (5 mice)/group. (*Figure 2—source data 2*) *p<0.05 vs. other groups. (**e**) Quantification of spine types. Mushroom, ANOVA, F(2,33)=24.7, p<0.0001*; Stubby, ANOVA, F(2,33)=25.4, p<0.0001*; Thin, ANOVA, F(2,33)=7.66, P=0.002*; power=0.9–1; n=12 neurons (6 mice) /group. *p<0.05 vs. other groups. (*Figure 2—source data 3*) (**f**) Z-stack images, pub and post-pub hippocampus, showing the effects of pubertal vehicle or memantine (MEM) treatment, a NMDAR blocker which does not alter NMDAR expression (*Cole et al., 2013*). Scale, 6 μm. (**g**) Averaged spine density. *Proximal: ANOVA, F(2,54)=64.12, p<0.0001*, power=1, n=17–20 neurons (4–5 mice) /group; Distal: ANOVA, F(2,56)=33.2, p<0.0001*, power=1, n=19–20 neurons (4–5 mice) /group. (*Figure 2—source data 4*) *p<0.05 vs. other groups. (**h**) Quantification of spine types. Mushroom, ANOVA, F(2,45)=89.9, p<0.0001*; Stubby, ANOVA, F(2,45)=9.4, P=0.0004*; Thin, ANOVA, F(2,45)=13.7, P=0.0001*; Bifurcated, ANOVA, F(2,45)=17.7, p<0.0001*; power=1, n=16 neurons (4–5 mice)/group. (*Figure 2—source data 5*) *p<0.05 vs. other groups.

The following source data is available for figure 2:

**Source data 1.** *Figure 2b*: NMDA EPSC/ AMPA EPSC ratios recorded from CA1 hippocampal pyramidal cells using whole cell patch clamp techniques for post-pubertal WT (a), α4 KO mice (b), WT hippocampus with SR95531 (c) and WT hippocampus with L-655,708 (L655) (d).

**Source data 2.** *Figure 2d*: Spine counts/20 μm on dendrites of CA1 hippocampal pyramidal cells – proximal (left) and distal (right) for pubertal (Pub), Post-pubertal (Post-pub) – vehicle (VEH), and Post-pub MK-801 (treated with MK-801 during the pubertal period).

**Source data 3.** *Figure 2e*: Spine counts/10 μm for different spine-types on dendrites of CA1 hippocampal pyramidal cells for *Figure 1c* for Pub, Post-pub vehicle (VEH) and Post-pub MK-801 (treated with MK-801during the pubertal period).

**Source data 4.** *Figure 2g*: Spine counts/20 μm on dendrites of CA1 hippocampal pyramidal cells – proximal (left) and distal (right) for α4 KO: pubertal (Pub), Post-pubertal (Post-pub) – vehicle (VEH), and Post-pub memantine (treated with memantine during the pubertal period).

**Source data 5.** *Figure 2h*: Spine counts/10 μm for different spine-types on dendrites of CA1 hippocampal pyramidal cells for *Figure 1c* for α4 KO: Pub, Post-pub vehicle (VEH) and Post-pub memantine (treated with memantine during the pubertal period).

## Spine density after alterations in pubertal NMDAR expression

NMDA-generated current was reduced at puberty (*Figure 2a,b*), even after blockade of GABARs at inhibitory synapses with 200 nM SR95531 (*Stell and Mody, 2002*). In contrast, NMDAR current was robust in the α4 KO hippocampus (*Figure 2a,b*), as we have previously shown for the δ KO hippocampus (*Shen et al., 2010*), and after total GABAR blockade with 120 μM SR95531 (*Figure 2a,b*). Thus, we tested the hypothesis that α4βδ receptors reduce spine number via impairment of NMDAR activation by examining whether increasing expression of NMDARs during puberty could reduce pruning by overwhelming the α4βδ-generated inhibition. To this end, MK-801 was administered, at a dose shown to increase hippocampal NMDAR expression compensatorily (*Gao and Tamminga, 1995*), during the pubertal period (0.25 mg/kg,i.p., once daily for 10 days). Spine density was evaluated post-pubertally. Pubertal MK-801 administration increased spine density by 100% in both proximal and distal dendrites post-pubertally (p<0.001). This reflected a significant increase in the density of mushroom and stubby spines (*Figure 2c–e*). Conversely, blockade of NMDARs with memantine, which does not alter NMDAR expression (*Cole et al., 2013*), administered to pubertal α4 KO mice reduced spine density post-pubertally by 50% (p<0.01). Memantine decreased mushroom and stubby spines and increased thin spines (p<0.05, *Figure 2f–h*). These data suggest that reduced NMDAR activity mediates synaptic pruning at puberty and that α4βδ GABARs are novel regulators of NMDARs which trigger this process.

## Spine density after blockade of α5βγ2 GABARs

In contrast to α4βδ GABARs, α5βγ2, the primary extrasynaptic GABAR in CA1 hippocampus (*Caraiscos et al., 2004*), did not impair NMDAR activation during puberty. Reduction of current through α5βγ2 GABARs was accomplished with the partial inverse agonist L-655,708 (L655, 50 nM) (*Ramerstorfer et al., 2010*), which had no effect on evoked NMDA current (*Figure 2a,b*) recorded from pubertal slices. In order to test the effect of GABAR sub-types on synaptic pruning, we

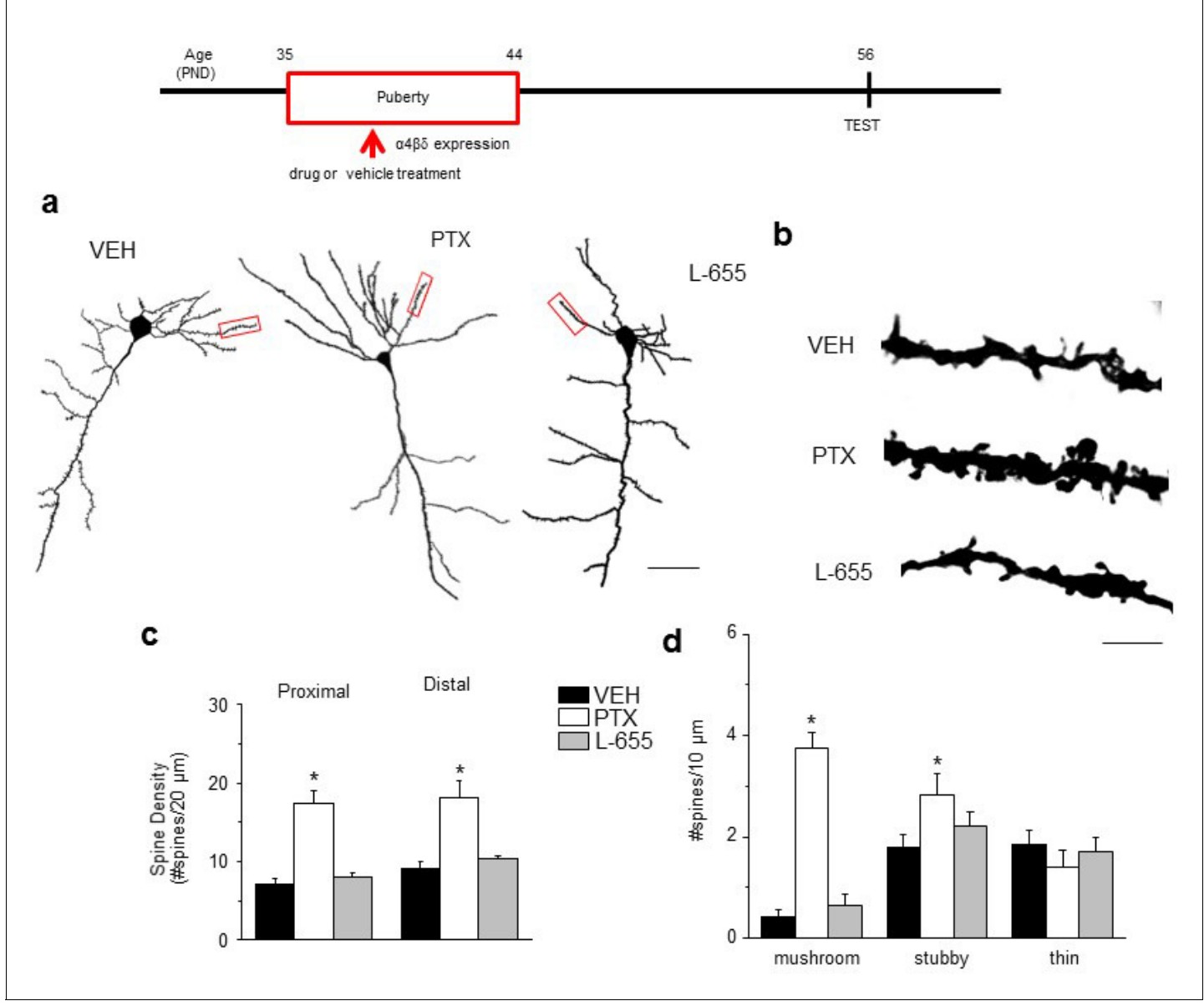

**Figure 3.** Effect of GABAR blockade on spine density in the post-pubertal hippocampus. Inset, Drug treatment during puberty (PND 35–44) was tested for its effect on post-pubertal spine density (PND 56). Drugs: PTX, picrotoxin, a GABAR antagonist; L655, L-655,708, an inverse agonist at α5-GABAR; VEH, vehicle (oil). (a) Neurolucida images, post-Pub CA1 pyramidal cells, following pubertal drug treatment; scale, 50 μm. (b) z-stack (100x) images; scale, 10 μm. (c) Spine density, Proximal (left): ANOVA, $F_{(2,30)}=45.5$, $p<0.0001*$, power=1; Distal (right): ANOVA, $F_{(2,30)}=60.8$, $p<0.0001*$, power=1; n=11 neurons (6 mice)/group. (*Figure 3—source data 1*) *$p<0.05$ vs. other groups. (d) Spine morphology changes. Mushroom, ANOVA, $F_{(2,45)}=104.2$, $p<0.0001*$; Stubby, ANOVA, $F_{(2,45)}=4.78$, $p=0.013*$; Thin, ANOVA, $F_{(2,45)}=1.37$, $P=0.27$; power=0.8–1, n=16 neurons (6 mice)/group. (*Figure 3—source data 1*) *$p<0.05$ vs. other groups. Lorazepam effects on spine density are depicted in *Figure 3—figure supplement 1*.

The following source data and figure supplements are available for figure 3:

**Source data 1.** *Figure 3c*: Spine counts/20 μm on dendrites of CA1 hippocampal pyramidal cells – proximal (left) and distal (right) for Post-pubertal (Post-pub) mice treated with L-655,708 (L655, left), vehicle (VEH, middle) or picrotoxin (Picro, right), during the pubertal period.

**Source data 2.** *Figure 3d*: Spine counts/10 μm for different spine-types on dendrites of CA1 hippocampal pyramidal cells for Post-pubertal (Post-pub) mice treated with L-655,708 (L655, left), vehicle (VEH, middle) or picrotoxin (Picro, right), during the pubertal period. Spines were identified as: mushroom, stubby, or thin.

**Figure supplement 1.** Pubertal lorazepam treatment does not alter spine density in post-pubertal mice.

*Figure 3 continued on next page*

*Figure 3 continued*

**Figure supplement 1—source data 1.** Spine counts/20 μm on dendrites of CA1 hippocampal pyramidal cells – proximal (left) and distal (right) for post-pubertal mice treated with MK-801 during the pubertal period.

administered L655 during the pubertal period (PND 35–44) and assessed spine density post-pubertally (PND 56). As predicted, pubertal administration of L655 produced no change in spine density post-pubertally (*Figure 3*), nor did the benzodiazepine lorazepam (*Figure 3—figure supplement 1*) which targets the primarily synaptic γ2-containing GABARs (*Sigel, 2002*). These findings suggest that α4βδ GABARs selectively reduce spine density at puberty. As expected, total GABAR blockade during puberty (picrotoxin, 3 mg/kg, i.p.) prevented pruning post-pubertally (*Figure 3*), increasing spine density by ~200%. Both mushroom (>900%) and stubby (100%) spines were increased while thin spines were decreased (75%, $p < 0.05$).

## Alterations in Kal7 expression across the pubertal period

Kal7 is a spine protein involved in dynamic spine changes (*Ma et al., 2008*). Kal7 expression in hippocampal dendrites decreased by almost 50% at puberty ($p < 0.05$) compared to pre-puberty (PND 28–30), an effect prevented by knock-out of α4, which increased its expression by ~120% ($p < 0.001$) (*Figure 4a–d*, *Figure 4—figure supplement 1*). We tested whether NMDARs played a role in Kal7 expression, which is activity-dependent (*Ma et al., 2011*). Increasing NMDAR expression at puberty increased Kal7 expression by ~100% ($p < 0.001$; *Figure 4e,f*), while NMDAR blockade with memantine (10 mg/kg, i.p.) reduced Kal7 expression by ~50% in the adult CA1 hippocampus (*Figure 4g,h*). These findings suggest that activity-dependent expression of Kal7 requires NMDAR activation that is regulated by α4βδ-mediated inhibition.

## Spine density changes in the Kal7 KO hippocampus

Because Kal7 is necessary for maintenance of spine number (*Ma et al., 2003*), we tested the hypothesis that synaptic pruning may be mediated by the decrease in Kal7 expression at the onset of puberty. To this end, we examined spine density in pubertal and post-pubertal hippocampus from the Kal7 KO mouse, where such a pubertal decrease could not occur. (Mice with both alleles of the Kal7 gene (*Kalrn7*) inactivated are referred to here as Kal7 KO.) In fact, synaptic pruning was prevented in the Kal7 KO, for which spine density was reduced by 40% for both age groups (*Figure 4i, j*). These data suggest that synaptic pruning may require the decrease in Kal7 expression at puberty in the WT mouse.

## Spine density and synaptic plasticity

Theoretical analysis has suggested that high densities of mature mushroom spines impair changes in synaptic strength (*Ruppin, 1999*), thus predicting that long-term depression (LTD) would be impaired by the high density of mature spines in post-pubertal α4 KO hippocampus. This was the case, where an $8 \pm 2.1\%$ depression was observed 1 hr after low frequency stimulation (LFS) compared to the $32 \pm 8.4\%$ depression observed in post-pubertal WT hippocampus ($p < 0.05$, *Figure 5a*). In contrast, theta burst-induction of NMDAR-dependent long-term potentiation (LTP), an in vitro model of learning, was not altered in α4 KO post-pubertal hippocampus (*Figure 5b*).

## Effects of spine density changes on spatial learning and re-learning

The behavioral outcome of altered spine density was tested by examining learning and re-learning using the hippocampus-dependent active place avoidance (APA) (*Pastalkova, 2006*) task and the multiple placement object relocation task (MPORT) (*Barker and Warburton, 2011*). In both tasks, post-pubertal α4 KO mice showed impaired acquisition and retention of the second location, despite initial learning scores similar to WT mice (*Figure 5c-d, f*). This was a cognitive deficit because locomotor activity, shock tolerance and object interest did not differ from WT (*Figure 5e,g*). As predicted, reducing synaptic pruning in the WT mouse with MK-801 impaired re-learning performance on MPORT, while restoring synaptic pruning in the α4 KO mice by blocking NMDARs with memantine, improved performance on this task (*Figure 5—figure supplement 1*). These data suggest that optimal cognition in adulthood is dependent upon adequate synaptic pruning in adolescence.

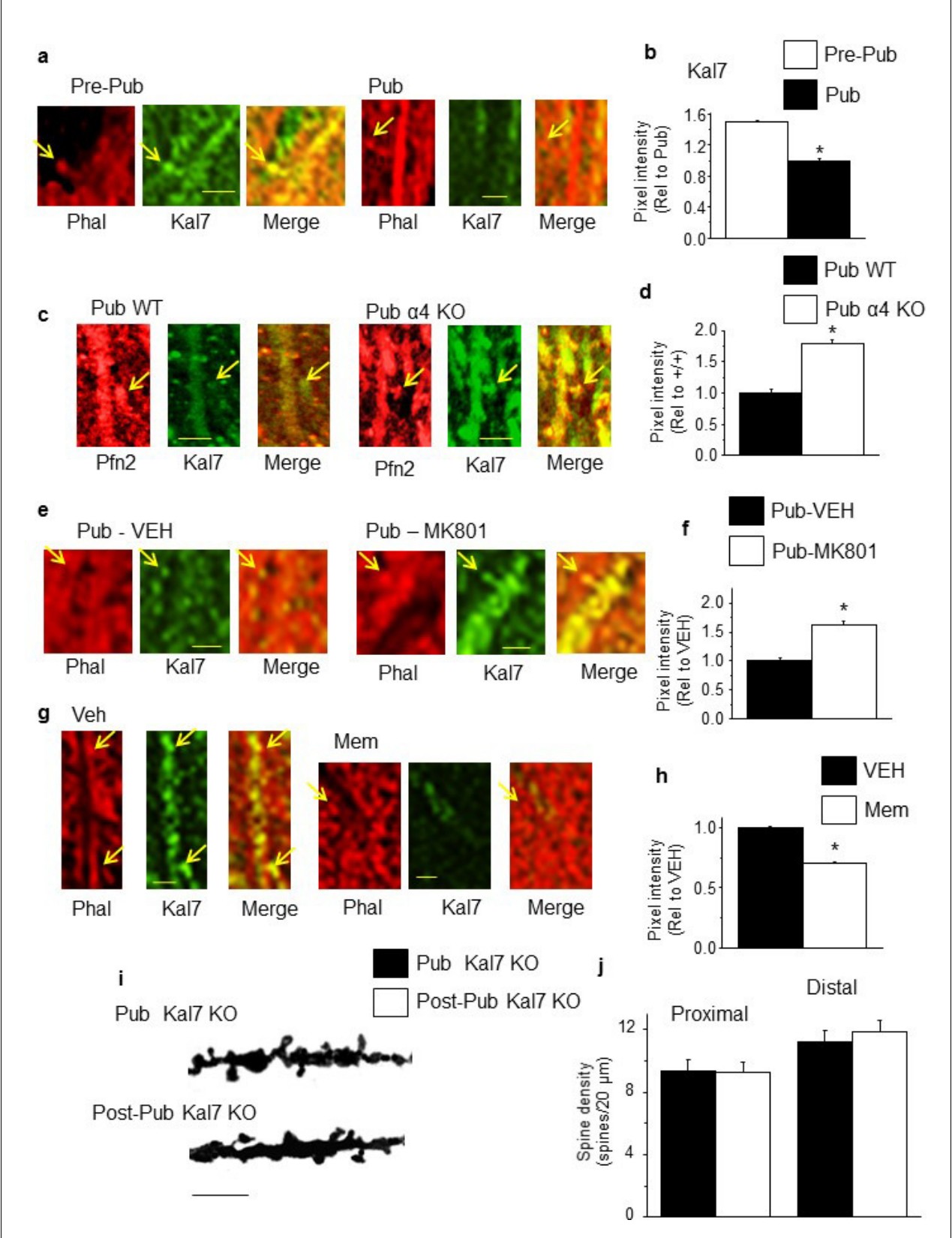

**Figure 4.** NMDA receptor-dependent Kalirin-7 expression decreases at puberty. (a,c,e,g) Representative images, scale, 2.5 µm. Arrows, spines. (a) Phalloidin (Phal), Kalirin-7 (Kal7) and merged images from pre-pub and pub CA1 hippocampus. (b) Mean pixel intensity, *t-test, t(26)=29.2, p<0.0001*,
*Figure 4 continued on next page*

*Figure 4 continued*

power=1; n=14 neurons (6 mice)/group. (c) Pfn2, Kal7 and merged images from pub WT and α4 KO CA1. (d) Mean pixel intensity, *t-test, t(26)=12.0, p<0.0001*, power=1; n=14 neurons (4 mice)/group. (e) Phal, Kal7 and merged images from pub CA1 hippocampus following in vivo treatment with vehicle or MK801 to increase NMDAR expression (*Gao and Tamminga, 1995*). (f) Mean pixel intensity, *t-test, t(26)=6.25, p<0.0001*, power=1; n=14 neurons (5 mice)/group. (g) Phal, Kal7 and merged images from post-pub CA1 hippocampus following in vivo treatment with vehicle or memantine (MEM), an NMDAR blocker. (h) Mean pixel intensity, *t-test, t(26)=6.5, p<0.0001*, power=1; n=14 neurons (5 mice)/group. Original uncropped images of Kal7 immunohistochemistry are shown in *Figure 4—figure supplement 1*. (*Figure 4—source data 1*) (i) Representative z-stack images, Pub, post-Pub Kal7 KO. Scale, 10 μm. (j) Averaged data, spine density. Proximal: t(32)=0.06, p=0.95, n=17 neurons (6 mice)/group; Distal: t(32)=0, p=1, n=17 neurons (6 mice)/group. (*Figure 4—source data 2*)

The following source data and figure supplement are available for figure 4:

**Source data 1.** *Figure 4b,d,f,h*: Measurements of Kalirin-7 (Kal7) luminescence taken from CA1 hippocampal pyramidal cells for Pre-pub and Pub WT (4b), Pub, WT and α4 KO (4d), Pub WT-treated with MK-801 or vehicle (VEH) (4f) and Post-pub WT-treated with memantine or VEH (4h).

**Source data 2.** *Figure 4j*: Spine counts/20 μm on dendrites of CA1 hippocampal pyramidal cells – proximal (left) and distal (right) for pubertal (Pub) and post-pubertal (Post-pub) Kal7 KO mice.

**Figure supplement 1.** Kalirin-7 expression varies across pubertal stage, α4 genotype and level of pubertal NMDAR expression.

## Effects of spatial learning on spine density

Our data suggest that when adolescent synaptic pruning is prevented, resulting in abnormally high spine density in the adult (α4 knock-out, MK-801 administration), re-learning is impaired. In order to explain this outcome, we examined the effect of initial learning and re-learning of MPORT on the distribution of spine types in CA1 hippocampal pyramidal cells. To this end, mouse brains were processed with the Golgi method by 1–2 hr after each learning or re-learning paradigm. Initial learning resulted in a ~150% increase in mushroom spine-types (p<0.05), with similar increases in stubby spine types (~200%, p<0.05) (*Figure 6*). After the second learning trial (re-learning), mushroom spine types were additionally increased by ~50–100% (p<0.05), while thin spine types were decreased by ~70% (*Figure 6*); the density of stubby spines did not change significantly. Thus, increases in mushroom spine density accompanied both learning and re-learning trials. Pubertal administration of picrotoxin (3 mg/kg, i.p.) to block GABARs also prevented synaptic pruning (*Figure 3*) and impaired re-learning (*Figure 5—figure supplement 2*) in the post-pubertal mouse. Under these conditions, mushroom spine density was almost 300% greater in the naïve condition compared to the untreated mouse (*Figure 3*, *6*), and additional increases in mushroom spine density were only observed after learning (~50% increase, p<0.05), but not after relearning (*Figure 6*). Stubby spines were additionally increased to a lesser extent (~20%) in the distal dendrites after learning. Because re-learning was impaired in this unpruned condition, these findings suggest that only successful learning/re-learning increases the mushroom spine density of CA1 hippocampal pyramidal neurons.

## Discussion

Pubertal synaptic pruning is seen in many CNS areas (*Huttenlocher, 1979*; *Zehr et al., 2006*; *Petanjek et al., 2011*), and is correlated with EEG changes in humans (*Campbell et al., 2012*), where the steepest reduction in slow wave (delta, 1–4 Hz) activity occurs at the onset of puberty. Our findings suggest that α4βδ GABARs play a critical role in this process. α4βδ GABARs are uniquely localized to spines during puberty where they reduce NMDAR current (*Shen et al., 2010*), necessary for spine stability (*Alvarez et al., 2007*). In contrast, blockade of the predominant extrasynaptic GABAR in CA1 hippocampus, α5β3γ2, did not facilitate NMDAR activation or alter synaptic pruning during the pubertal period, nor did modulation of synaptic γ2-containing GABARs with 200 nM SR95531 or lorazepam. Thus, these findings suggest that α4βδ GABARs selectively trigger synaptic pruning at puberty via impairment of NMDAR activation. This is in contrast to visual cortex, where synaptic α1β2γ2 GABARs target dendritic spines (*Kawaguchi and Kubota, 1997*), and play a role in spine maturation (*Heinen et al., 2003*). Pubertal synaptic pruning is not dependent upon the ovarian hormone estradiol (*Yildirim et al., 2008*), which in fact increases spine density (*Woolley et al., 1997*).

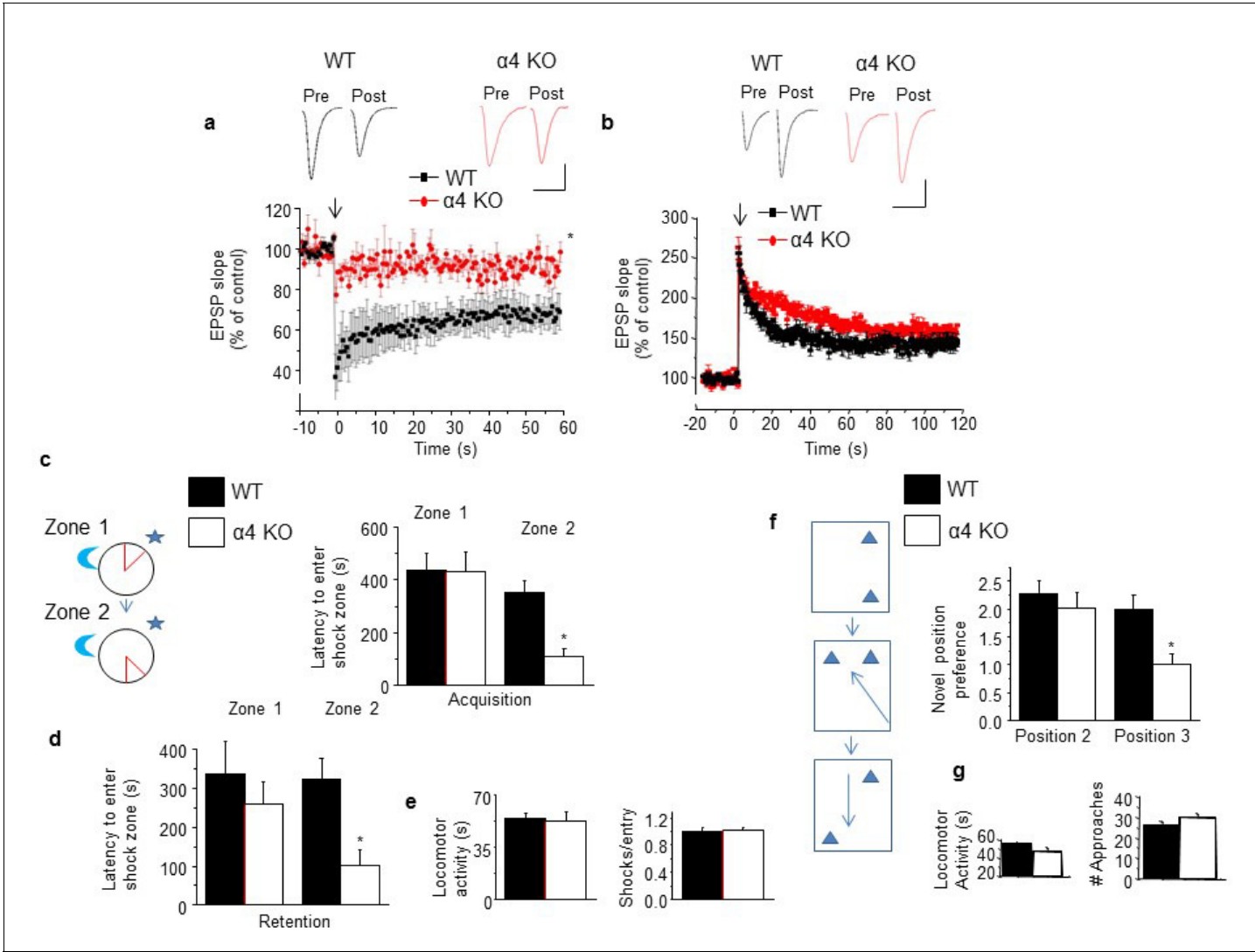

**Figure 5.** Induction of long-term depression and re-learning are impaired under conditions of high spine density in the α4 KO mouse. (a) Induction of long-term depression (LTD) using low frequency stimulation (arrow). WT, black, α4 KO, red. *t-test, t(6)=3.56, p=0.01, power=0.84; n=4/group. (*Figure 5—source data 1*) Inset, representative field EPSPs. Scale, 0.2 mV, 20 ms. (b) Induction of long-term potentiation (LTP) using theta burst stimulation (arrow). WT, black, α4 KO, red. t-test, t(7)=0.28, p=0.78; n=4–5/group. (*Figure 5—source data 2*) Inset, representative field EPSPs. Scale, 0.2 mV, 25 ms. (c) [Inset, The active place avoidance task (APA). The animal is trained to avoid a shock zone (red) on a rotating arena. Day 1, training for zone 1; day 2, training for zone 2.] Average latency to enter shock zone 1 (Z1) and 2 (Z2), Acquisition. *t-test, Zone 1, t(9)=0.02, p=0.99; Zone 2, t(10) =3.37, p=0.007*, power=0.86; n=5–7 mice. (d) Average latency to enter shock zone 1 (Z1) and 2 (Z2), Retention. * t-test, Zone 1, t(9)=1.17, p=0.27; Zone 2, t(10)=3.08, p=0.012*, power=0.80; n=5–7 mice. (*Figure 5—source data 3*) (e) Locomotor activity (left, t test, t(10)=0.67, p=0.52) and # shocks/entry, a measure of escape behavior (right, t test, t(10)=0.08, p=0.93). n=5–7 mice/group. (*Figure 5—source data 4*) (f) Inset, the multiple placement object recognition task (MPORT). Sequence of positions (1–3) of object 2 across 3 training trials. Novel position preference for positions 2 and 3. Position 2, *t-test, t(23)=0.85, p=0.40; Position 3, t(23)=4.61, p<0.0001*, power=1; WT, n=15 mice; α4 KO, n=10 mice. (*Figure 5—source data 5*) (g) Locomotor activity (left, t-test, t(23)=0.34, p=0.74; WT, n=15 mice; α4 KO, n=10 mice) and # approaches, a measure of object interest (right, t t-test, t(23)=0.97, p=0.339; WT, n=15 mice; α4 KO, n=10 mice) (*Figure 5—source data 6*). Effects on MK-801 and memantine on learning and re-learning are depicted in *Figure 5—figure supplement 1*. Picrotoxin effects on learning and re-learning are depicted in *Figure 5—figure supplement 2*.

The following source data and figure supplements are available for figure 5:

**Source data 1.** *Figure 5a*: Percent baseline slope of field EPSPs recorded after low frequency (1 Hz) stimulation to induce LTD for post-pubertal WT and α4 KO CA1 hippocampus (120 min, 30 s intervals).

**Source data 2.** *Figure 5b*: Left, Percent baseline slope of field EPSPs recorded after theta burst stimulation to induce LTP for post-pubertal WT and α4 KO CA1 hippocampus (final 20 min 100 min after LTP induction, 30 s intervals).

*Figure 5 continued on next page*

*Figure 5 continued*

**Source data 3.** *Figure 5c*: Learning acquisition (left) and retention (right) for zone 1 of the active place avoidance task (APA).

**Source data 4.** *Figure 5e*: #shocks/entry (left) and locomotor activity (right) for post-pub WT and α4 KO mice assessed for the active place avoidance task (APA).

**Source data 5.** *Figure 5f*: Learning acquisition for zones 1–3 for post-pub WT (left) and α4 KO (right) of the multiple placement object relocation task (MPORT).

**Source data 6.** *Figure 5g*: Locomotor activity (left) and # approached, a measure of interest (right) for for post-pub WT (left) and α4 KO (right) mice using MPORT.

**Figure supplement 1.** NMDAR antagonist treatment alters behavioral flexibility.

**Figure supplement 1—source data 1.** Learning acquisition for positions (Pos) 2 and 3 for the multiple placement object relocation task (MPORT).

**Figure supplement 1—source data 2.** Learning acquisition for positions (Pos) 2 and 3 for the multiple placement object relocation task (MPORT).

**Figure supplement 2.** Pubertal GABAR antagonist treatment impairs behavioral flexibility post-pubertally.

**Figure supplement 2—source data 1.** Learning acquisition for positions (Pos)1–3 for the multiple placement object relocation task (MPORT).

In the present study, MK-801 administration during the pubertal period of female mice increased the density of dendritic spines post-pubertally, resulting in increases in both mushroom and stubby spine-types. A recent study (*Han et al., 2013*) reports that similar MK-801 treatment across the peri-pubertal period in male rats also increases the density of stubby spines in hippocampus, but decreases the mushroom spine-types. The reason for the disparity in the two outcomes may be due to gender or developmental differences, as the latter study used rats across an age range that likely includes both pre-pubertal and pubertal ages (*Marty et al., 2001*).

Although MK-801 is an NMDA antagonist, it has been shown to increase NMDAR expression, as a compensatory effect 24 hr after administration (0.1–1.0 mg/kg), selectively in the CA1 region of the hippocampus (*Gao and Tamminga, 1995*). This is the most likely mechanism by which MK-801 increased spine density in the present study where injections were administered once a day during the pubertal period. In the pre-frontal cortex, only a single low dose of MK-801 increases NMDAR expression on pyramidal cells (*Xi et al., 2009*), while higher doses have no effect, but decrease NMDAR expression and AMPA receptor-mediated currents of fast-spiking interneurons (*Wang and Gao, 2012*) suggesting that effects of this drug are cell-type and region specific. Other studies have suggested that peri-adolescent administration of MK-801 alters GABAergic circuits in adulthood (*Thomases et al., 2013*). We cannot rule out the additional possibility that the MK-801 treatment also produced changes in the GABAergic circuitry of the hippocampus which may have contributed to the observed changes in spine density post-pubertally.

α4βδ GABAR-induced impairment of NMDAR activation at puberty reduces expression of Kal7, a spine protein involved in spine restructuring which binds to the post-synaptic density (PSD) (*Penzes et al., 2001*). Kal7 activates the small GTPase Rac1 which regulates the actin cytoskeleton via P21-activated kinases within the spine (*Ma et al., 2014*). Although many proteins localize to the spine, Kal7 is one of the few shown to be necessary for maintenance of existing spines (*Penzes et al., 2001*; *Ma et al., 2003*). That may explain results from the present study where a 50% reduction in Kal7 expression at puberty resulted in a 50% decrease in spine number. The fact that synaptic pruning was prevented in the absence of Kal7 expression suggests that dynamic regulation of Kal7 expression during adolescence may be one factor underlying synaptic pruning. However, many other spine proteins have been identified which alter spine density, including shank3, IQGAP1, valosin, axin and NEDD9 (*Roussignol et al., 2005*; *Gao et al., 2011*; *Wiesmann et al., 1975*; *Knutson et al., 2016*; *Chen et al., 2015*). Thus, synaptic remodeling during adolescence may incorporate a more complex array of spine protein changes.

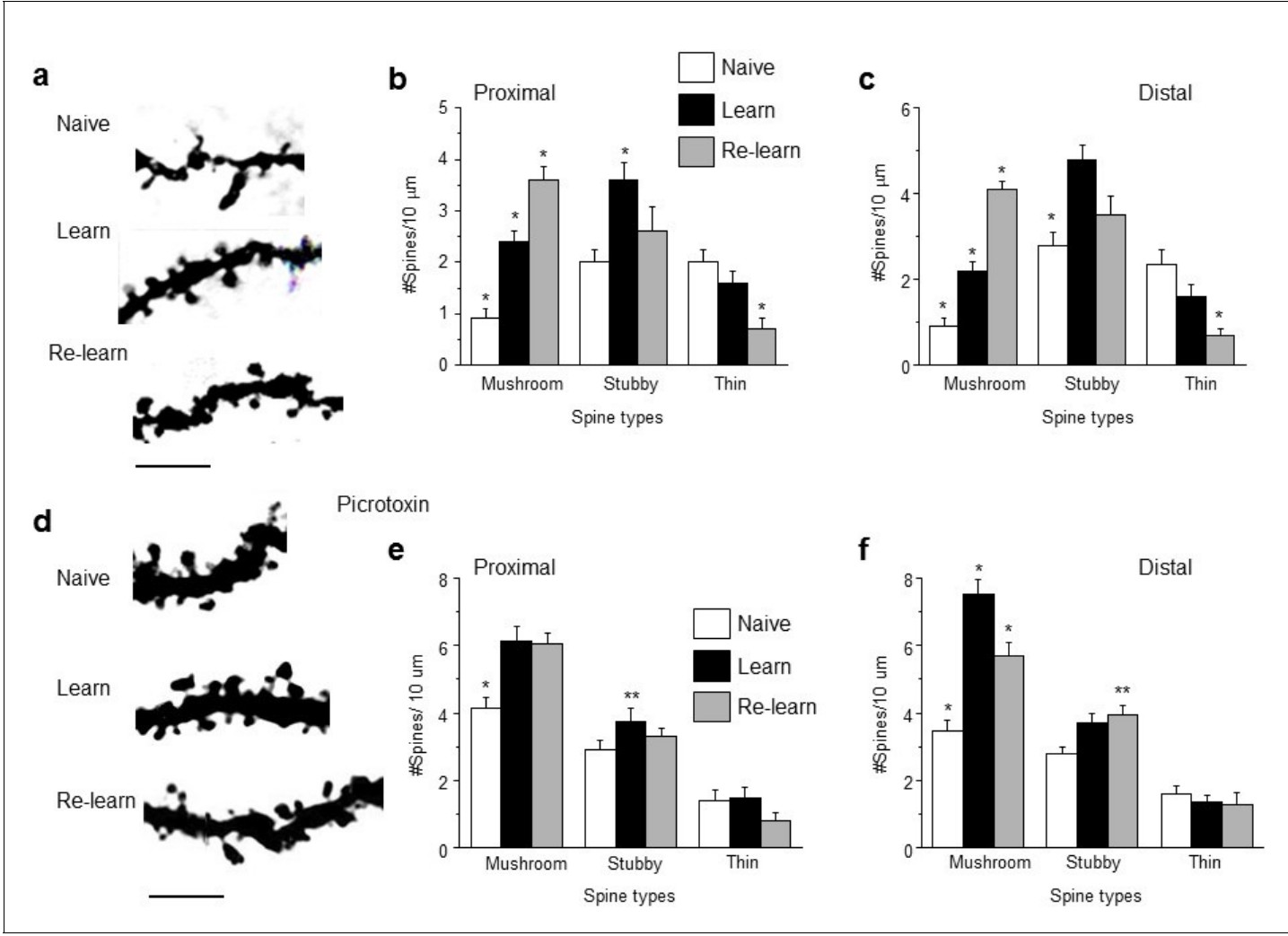

**Figure 6.** Learning and re-learning increase mushroom-type dendritic spines in CA1 hippocampus following adolescent synaptic pruning. (a) Representative z-stack images from CA1 hippocampal pyramidal cells illustrating changes in spine type and number after hippocampal-dependent learning and re-learning, compared to naïve conditions. Scale, 5 μm. (b, c) Means ± S.E.M. for proximal and distal dendrites. Proximal, Mushroom, ANOVA, $F_{(2,39)}=44.9$, $p<0.0001*$, power=1; Stubby, ANOVA, $F_{(2,39)}=6.0$, $p=0.005*$. power=0.86; Thin, ANOVA, $F_{(2,39)}=7.24$, $p=0.004*$; power=0.97, n=14 neurons (5 mice)/group. $*p<0.05$ vs. other groups. Distal, Mushroom, ANOVA, $F_{(2,39)}=84.1$, $p<0.0001*$, power=1; Stubby, ANOVA, $F_{(2,39)}=13.7$, $p<0.0001*$, power=1; Thin, ANOVA, $F_{(2,39)}=13$, $p<0.0001*$; power=1, n=14 neurons (4–6 mice)/group. $*p<0.05$ vs. other groups. $**p<0.05$ vs. naïve. (*Figure 6—source data 1*) (d) Representative z-stack images from hippocampus of adult mice treated during the pubertal period with 3 mg/kg picrotoxin (*Figure 3*) to prevent synaptic pruning. Changes in spine type and number are evident after hippocampal-dependent learning and re-learning, compared to naïve conditions. Scale, 5 μm. (e,f) Means ± S.E.M. for proximal and distal dendrites. Proximal, Mushroom, ANOVA, $F_{(2,39)}=12.6$, $p<0.0001*$, power=0.99; Stubby, ANOVA, $F_{(2,39)}=3.78$, $p=0.03*$. power=0.86; Thin, ANOVA, $F_{(2,39)}=0.87$, $p=0.43$, n=14 neurons (5 mice)/group. $*p<0.05$ vs. other groups. $**p<0.05$ vs. naïve. Distal, Mushroom, ANOVA, $F_{(2,39)}=33.1$, $p<0.0001*$, power=1; Stubby, ANOVA, $F_{(2,39)}=3.87$, $p<0.029*$, power=1; Thin, ANOVA, $F_{(2,39)}=0.42$, $p=0.66$, n=14 neurons (5 mice)/group. $*p<0.05$ vs. other groups. $**p<0.05$ vs. naïve. (*Figure 6—source data 2*)

The following source data is available for figure 6:

**Source data 1.** *Figure 6b*: (proximal), 6c (distal), Spine counts/10 μm for different spine-types on dendrites of CA1 hippocampal pyramidal cells assessed 1–2 hr after learning, re-learning or naïve conditions.

**Source data 2.** *Figure 6e*: (proximal), 6f (distal), Spine counts/10 μm for different spine-types on dendrites of CA1 hippocampal pyramidal cells assessed 1–2 hr after learning, re-learning or naïve conditions in mice with high spine densities (treated with picrotoxin during the pubertal period).

Several studies have suggested that scavenging by microglia (*Schafer et al., 2012*) or direct autophagy (*Tang et al., 2014*) prunes spines. In particular, the C4 complement system is abnormal in schizophrenia; C4 activates C3, which binds to target spines and promotes their engulfment by phagocytic cells (*Sekar et al., 2016*). This system may be target and developmentally specific – as it does not underlie adolescent pruning in CA1 hippocampus (*Shi et al., 2015*). In addition, the involvement of the complement cascade in synaptic pruning would likely be the final step in spine removal and does not preclude involvement of other systems, such as NMDAR inhibition by α4βδ receptors which would be the initial trigger for the pruning process.

The mushroom spines were selectively removed during pubertal synaptic pruning in the wild-type mouse, leaving an abundance of thin spines. In order to better understand the role of the various spine types in cognition, we directly examined the changes in spine morphology produced by learning and re-learning a spatial task in the wild-type mouse. Our results suggest that learning increased mushroom spines, consistent with earlier reports (*Beltrán-Campos et al., 2011*; *González-Ramírez et al., 2014*), which we also observed by 2 hr after re-learning, when the density of thin spines also decreased. These two spine types are well-characterized and sub-serve different functions: Thin spines express abundant numbers of NMDARs, are highly motile and plastic (*Kasai et al., 2003*; *Holtmaat et al., 2006*), while the larger mushroom spines are relatively more stable and express AMPA receptors predominantly (*Dumitriu et al., 2010*; *Matsuzaki et al., 2001*). Recent studies using two photon technology to assess changes in spines produced by NMDAR-dependent LTP, an in vitro model of learning, have shown that smaller 'thin' spines develop enlarged heads and resemble 'mushroom' spines following LTP induction, accompanied by increased expression of AMPA receptors (*Hill and Zito, 2013*; *Kopec et al., 2006*; *Harvey and Svoboda, 2007*). These larger spine heads have larger post-synaptic densities and active zones, consistent with stronger synaptic connections (*Bell et al., 2014*). Typically, these changes are first observed within 15 min. with peak effects 40 min to 2 hr after LTP induction, similar to the time-course of our effects.

Our findings suggest that a high spine density produces cognitive impairment. Although modelling studies have predicted the cognitive outcome of increases in spine density (*Ruppin, 1999*), most experimental studies to date have only examined the effect of spine loss on cognition (*Oaks et al., 2016*). The high spine density in the unpruned condition in the present study selectively impaired re-learning, while learning a spatial task was similar to controls. The impairments in re-learning in the unpruned conditions may be due to the higher prevalence of mushroom-type spines and lower density of thin spine-types. Mushroom spine density was only increased above its already elevated level in the unpruned mouse following the learning trial. No additional increases in mushroom spines were observed for the unsuccessful re-learning trial. There may be a maximum number of mushroom spines that can be supported by a dendritic segment due to the energy requirement necessary for spine maintenance or due to spatial constraints. Alternatively, the lower density of thin spines may have been insufficient to permit successful re-learning.

The changes in spine density observed after learning and re-learning protocols were of the same magnitude as those observed during adolescence, reflecting the high plasticity of spines, although changes occurred in the opposite direction. Changes in spine types were distinct for the two events as well: during learning and re-learning, mushroom spine-types increased, while these spine-types decreased during adolescence. In addition, the spine density of the control, post-pubertal mice may reflect the lower limit of synapse number because these mice were housed in non-enriched cages with limited sensory stimuli. Numerous studies have shown that exposure to more complex environments, even for 1 hr per day, increases spine density in many areas of the CNS, including CA1 hippocampus (*Turner and Greenough, 1985*; *Jung and Herms, 2014*; *Kitanishi et al., 2009*). Recent studies report that environmental enrichment increases large spines and enhances spine turnover (*Jung and Herms, 2014*; *Kitanishi et al., 2009*), suggesting that spine dynamics are environmentally specific.

LTD induction was also impaired in the unpruned condition (α4 KO), consistent with other reports which describe impairment in LTD under conditions where thin spine number is reduced (*Spiga et al., 2014*). Two photon studies have indeed shown that this protocol reduces thin spine number (*Nägerl et al., 2004*), thus suggesting that a critical number of thin spines may be necessary for LTD induction. LTD has been suggested as a cellular mechanism underlying synaptic changes necessary for re-learning (*Nicholls et al., 2008*), where weaker synaptic connections mediated by thin spines are reduced. In contrast to the present results, studies reporting more massive decreases

in thin spine number, as a result of aging or chronic alcohol exposure, also observed impairments in learning and LTP (*Dumitriu et al., 2010*; *Spiga et al., 2014*), suggesting that the critical number of thin spines necessary for learning is below that required for relearning.

The results from the present study may have relevance for the cognitive impairment in autism and schizophrenia where synaptic pruning is abnormal (*Hutsler and Zhang, 2010*; *van Spronsen and Hoogenraad, 2010*). Adolescent synaptic pruning of the temporal lobe does not occur in autism (*Tang et al., 2014*), leaving an abundance of dendritic spines (*Hutsler and Zhang, 2010*) which are associated with impairments in reversal learning (*D'Cruz et al., 2013*), similar to our results in the α4 KO mouse. In fact, reduced α4 expression has been reported in autism (*Fatemi et al., 2010*), which is correlated with increased risk of developing this disorder (*Collins et al., 2006*), although identified genetic abnormalities in α4 and/or δ genes in autism and schizophrenia are relatively rare (*Ma et al., 2005*; *Bullock et al., 2008*). Both disorders are more likely to occur in males (*Croen et al., 2002*; *Iacono and Beiser, 1992*), for which we also show α4βδ involvement in adolescent pruning. Initial deficits of autism appear in early childhood, but loss of cognitive gains are frequently reported in adolescence following improvement earlier in development (*Sigman and McGovern, 2005*; *Gillberg and Steffenburg, 1987*). Thus, the lack of synaptic pruning in adolescence may contribute to this developmental slow-down. The results from the present study may suggest novel therapeutic strategies to normalize disrupted synaptic pruning in these disorders.

## Materials and methods

### Animal subjects

Female and male C57/BL6 mice were housed in a reverse light:dark cycle (12: 12). Mice were tested for spine density at puberty onset (~PND 35) to compare with a post-pubertal age (PND 56). In some cases, pubertal mice (~PND 35–44) were injected with drugs or vehicle (oil) to target certain populations of GABARs or NMDARs and tested for spine density and learning/re-learning at 8-weeks of age. This pubertal time period was selected because it has been established that α4βδ GABARs increase expression on dendritic spines of CA1 hippocampal pyramidal cells beginning at puberty onset (vaginal opening or preputial separation, ~PND 35) and are maintained for the following 9 days (*Shen et al., 2010*; *Aoki et al., 2012*). In one study pre-pubertal mice (PND 28–31) were also tested.

In some studies, mice with deletions of the GABAR α4 subunit or kalirin-7 (Kal7) were used. α4 KO mice have mutations in exon 3 of *Gabra4* and were developed on a mixed C57BL/6J and SJL genetic background (*Chandra et al., 2006*) and back-crossed with C57BL/6J mice. Both sets of WT and α4 KO mice were bred on site from α4+/- mice originally supplied by G. Homanics (Univ. of Pittsburgh), with additional C57BL/6J mice from Jackson Laboratories (Bar Harbor, Maine) because results were similar to WT mice bred in-house. Genotyping of the tails was used to identify mice that were homozygous α4 KO. α4 KO mice are functional δ knock-outs (*Sabaliauskas et al., 2012*); they were used rather than δ KO to spare the α1βδ present on interneurons (*Glykys et al., 2007*). Kal7 KO mice were supplied by R.E. Mains (U. Conn. Health Center) (*Ma et al., 2008*). These mice lack the terminal exon unique to the Kal7 gene (*Kalrn7*) and were developed on a C57BL/6J background. Female mice were used because the onset of puberty is a physical sign (vaginal opening) that is directly correlated with the hormonal changes that trigger α4βδ GABAR expression, which has been well-characterized (*Shen et al., 2007*).

Drugs administered during puberty (once a day for 10 d – PND 35-PND 44): picrotoxin at a dose sub-threshold for seizure (*Verleye et al., 2008*; *Zolkowska et al., 2012*) (3 mg kg$^{-1}$, i.p.) to block all GABARs; L-655,708 (0.35 mg kg$^{-1}$, i.p.), an inverse agonist of α5-containing GABARs (*Ramerstorfer et al., 2010*; *Zurek et al., 2012*); MK-801 (0.25 mg kg$^{-1}$, i.p.), which at this dose, increases NMDAR expression (*Gao and Tamminga, 1995*); memantine (10 mg kg$^{-1}$, i.p.), an NMDAR antagonist which does not alter NMDAR expression (*Cole et al., 2013*), and lorazepam (0.25 mg kg$^{-1}$, i.p. in oil), which targets γ2-containing GABARs (*Sigel, 2002*). Unless otherwise indicated, saline was used as vehicle. Estrous cycle stage was determined by the vaginal cytology in 8-week old animals with established regular cycles, and these mice were not used in the stage of proestrus. Procedures were in accordance with the SUNY Downstate Institutional Animal Care and Use Committee.

## Golgi stain procedure

Whole brains from euthanized animals were processed for Golgi impregnation using the FD Neurotechnologies FD Rapid Golgi Stain kit. Coronal sections were prepared using a vibratome (Leica VT1200s) set to a thickness of 250 μm. Pyramidal cells from the CA1 hippocampus were reconstructed using Neurolucida software (MicroBrightField). The neurons were viewed with a 100× oil objective on an Olympus BX51 upright light microscope. The Neurolucida program projects the microscope image onto a computer drawing tablet. The neuron's processes are traced manually while the program records the coordinates of the tracing to create a digital, three-dimensional reconstruction. Z-stack projection photomicrographs (0.1 μm steps) were taken with a Nikon DS-U3 camera mounted on a Nikon Eclipse Ci-L microscope using a 100x oil objective and analyzed with NIS-Elements D 4.40.00 software. Camera Lucida drawings of dendrites were completed using a Nikon 710 microscope at 100x oil with a drawing tube attached.

## Spine density measurement

Reconstructed neurons were analyzed using Neurolucida Explorer built-in Sholl analysis software for spine density. Proximal dendrites were one-third of the distance or less from the cell soma while distal dendrites were one-third of the distance or less from the ends of dendritic branches. Spine density was similar in stratum oriens and stratum radiatum; therefore, these data were pooled. Spine types were determined using the semi-automated Spine Classifier of NeuronStudio (http://research.mssm.edu/cnic/tools-ns.html), a program that allows for the reconstruction of neurons and classification of spines from z-stacks. Briefly, stubby spines had a length to width ratio of ∼1, mushroom spines were identified by a $\geq$.35 μm head width, with a head dia:neck dia >2, while thin spines were classified if the head dia: neck dia < 1.2 and a length:width >3 (*Arellano et al., 2007*). All spine density and morphology assessments were made with the investigator blinded to the condition of the animals tested.

## Immunocytochemistry

Mice were anesthetized with urethane (0.1 ml 40%) and transcardially perfused using a peristaltic pump with a flow-rate of 12–15 mls/min, first with saline, followed by 4% paraformaldehyde (PFA) buffered to pH 7.4 with 0.1 M phosphate buffer (PB). Brains were dissected and post-fixed 48 hr in 4% PFA at 4°C. Coronal sections of the dorsal hippocampus were cut on a vibratome (Leica VT1200s) at a thickness of 35 μm. Sections were blocked in 0.01 M PBS supplemented with 1% bovine serum albumin, 0.25% Triton and 0.05% sodium azide for 2 hr. Then, sections were incubated with anti-Kal7 (ab52012, Abcam, 1:200) and, in some cases, anti-Pfn2 (60094-2-Ig, Proteintech, 1:50), to detect actin, diluted in the blocking solution overnight at 4°C. After washing, sections were incubated with fluorescent secondary antibody, or in some cases, fluorescent phalloidin, to detect actin: For staining using Kal7 and Pfn2, rabbit anti-goat Alexafluor 568 and donkey anti-mouse Alexafluor 488 (both at 1:500), respectively, were used. For staining using Kal7 and phalloidin, rabbit anti-goat Alexafluor 488 (1:500) and phalloidin-conjugated to Alexafluor 568 (1:20), respectively, were used. Following a 2 hr incubation at room temperature, sections were mounted on slides with ProLong Gold Antifade Reagent. Images were taken with a Olympus FluoView TM FV1000 confocal inverted microscope with objective UPLSAPO 60x NA:1:35 (Olympus, Tokyo, Japan) to show Kal7, Pfn2 or phalloidin and merged images. Images were analyzed for luminosity (Kal7 staining) using the region of interest (ROI) program of Image J software (NIH). In all experiments, actin is displayed as red and Kal7 as green. In order to enhance visualization of dendritic spines for *Figure 4*, the brightness is increased by 12 and the contrast by 40 in all images. However, the original non-enhanced images are presented in *Figure 4—figure supplement 1*.

## Hippocampal slice preparation

Mice were rapidly decapitated; the brains were removed and cooled using an ice cold solution of artificial cerebrospinal fluid (aCSF) containing (in mM): NaCl 124, KCl 2.5, $CaCl_2$ 2, $NaH_2PO_4$ 1.25, $MgSO_4$ 2, $NaHCO_3$ 26, and glucose 10, saturated with 95% $O_2$, 5% $CO_2$ and buffered to a pH of 7.4. Following sectioning at 400 μm on a Leica VT1000S vibratome, slices were incubated for 1 hr in oxygenated aCSF.

## Hippocampal slice voltage clamp electrophysiology

Pyramidal cells in the CA1 hippocampal slice were visualized using a differential interference contrast (DIC)-infrared upright microscope, and recorded using whole cell patch clamp procedures in voltage clamp mode at 2630° C, as described (*Shen et al., 2010*). Patch pipets were fabricated from borosilicate glass using a Flaming-Brown puller to yield open tip resistances of 2–4 MΩ. For whole cell recordings of miniature excitatory post-synaptic currents (mEPSCs), the aCSF contained 120 µM SR95531 (6-imino-3-(4-methyoxyphenyl)-1(6H)-pyridazinebutanoic acid hydrobromide) to block GABARs. (Pipet solution (in mM): 140 K-gluconate, 2 $MgCl_2$, 10 HEPES, 10 BAPTA, 2 Mg-ATP, 0.5 $CaCl_2$-H2O, 0.5 Li-GTP, pH 7.2, 290 mOsm.) Recordings were carried out at -60 mV. 1 µM tetrodotoxin (TTX) was added to block voltage-gated Na+ channels.

Recordings were conducted with a 2 kHz 4-pole Bessel filter at a 10 kHz sampling frequency using an Axopatch 200B amplifier and pClamp 9.2 software. Electrode capacitance and series resistance were monitored and compensated; access resistance was monitored throughout the experiment, and cells discarded if the access resistance increased more than 10% during the experiment. In most cases, the data represent one recording/animal.

## NMDA/AMPA EPSC ratio

Whole cell patch clamp recordings were carried out, as above, except that the aCSF contained 1 mM $MgCl_2$ (instead of 2 mM), 10 µM strychnine, 10 µM D-serine, and 50 µM CGP 35348, as previously described (*Shen et al., 2010*). Excitatory currents were evoked in the presence of 200 nM SR95531, in order to block the synaptic GABARs (*Stell and Mody, 2002*), with low frequency stimulation (0.05 Hz) at intensities close to threshold (100–400 µA) using a tungsten bipolar electrode placed ∼500 µm away in the stratum radiatum. Stimulation intensity was adjusted to achieve an EPSC amplitude of ∼400–500 pA (typically 75–150 µA). After baseline recordings of the glutamatergic EPSC, 5 µM NBQX was applied to unmask the NMDA component. In some cases, the NMDAR antagonist APV (50 µM) was applied to verify the nature of the NMDA current. The NMDA:AMPA ratio was calculated as (*amp. EPSPNMDA)/(amp. EPSPNMDA+AMPA) – (amp. EPSPNMDA)*. Currents were recorded from pubertal hippocampus, WT or α4 KO, to assess the role of α4βδ GABARs in reducing NMDAR current. In some cases L-655,708 (50 nM) or SR95531 (120 µM) was bath applied to block α5βγ2 GABARs or all GABARs, respectively.

## Data analysis

Evoked EPSCs and mEPSCs were detected using a threshold delimited event detection subroutine in pClamp10.3. Only data with a stable baseline and rapid rise time were included in the analysis. Event frequency was assessed and averaged.

## Long-term depression (LTD) and long-term potentiation (LTP)

The aCSF was similar to above except that the $MgSO_4$ concentration was 1 mM. Hippocampal slices were placed between nylon nets in a submerged chamber of an upright microscope. Field EPSPs (fEPSPs) were recorded extracellularly from the stratum radiatum of CA1 hippocampus using an aCSF-filled glass micropipet (1–5 mΩ) in response to stimulation of the Schaffer collateral-commissural pathway using a pair of insulated tungsten bipolar electrodes. The intensity of the stimulation was adjusted to produce 50% of the maximal response. LTD was induced using LFS (1 Hz) for 900 pulses (15 min) (*Dunwiddie and Lynch, 1978*). fEPSP slope was assessed every 30 sec with an Axoprobe-1A amplifier and pClamp 10.3 for 20 min. before and 1 hr after LTD induction. LTP was induced using theta burst stimulation (*Larson et al., 1986*) (TBS, 8–10 trains of 4 pulses at 100 Hz, delivered at 200 ms intervals, repeated 3x at 30s intervals) which is a physiological stimulation pattern (*Larson et al., 1986*). EPSP responses were recorded at 30s intervals for 20 min. before and 120 min. after TBS (producing 1–4 mV EPSPs). For both paradigms, the strength of synaptic excitatory responses was assessed by measuring the slope (initial 20–80%) of the EPSP rising phase. Data are expressed as a% of the average response from the 20 min. control period for each slice, and are averaged for all slices (mean ± SEM) across the time-course of the experiment, as we have described (*Shen et al., 2010*).

## Drugs

All drugs were from Sigma Chemical Co. (St. Louis, MO).

## Tests of spatial learning – Active place avoidance (APA) task

This is a hippocampal-dependent spatial memory task (*Koistinaho et al., 2001*), which requires LTP in the CA1 hippocampus (*Pastalkova, 2006*). After an initial 10 min habituation to a rotating platform (40 cm dia, 1 rpm), mice were trained for 3 10-min trials/hour to avoid a mild foot shock (<0.2 mA, sub-threshold for stress hormone release [*Friedman et al., 1967*]) in a 60° sector of the disk (Inset, *Figure 5*). The time to first enter the avoidance zone for each trial was assessed as an indicator of learning acquisition, and 120 s was set as the learning criterion (*Shen et al., 2010*). Additional trials were administered if the animals did not reach the learning criterion of a 120 s latency to first enter the avoidance zone. On day 2, animals were initially tested with the shock zone position from the previous day (first trial, zone 1) to reactivate their memory of the previous day. Then, the shock zone was changed to a different location (zone 2) and animals trained until learning criteria was achieved. On day 3, animals were tested for retention of zone 2. All trials and inter-trial intervals were 10 min long. The number of trials to reach learning criterion and the average latency to enter the shock zone (trial #3) were assessed as measures of learning acquisition for the initial location (zone 1) and re-learning of the second location (zone 2). In addition retention of this spatial memory was also assessed for both zone 1 and zone 2 and expressed as the latency to enter the shock zone for the first trial on the day after learning.

The position of the avoidance zone was stationary with respect to the room spatial frame of reference, which required active avoidance behavior because the disk was rotating. The position of the mouse on the disk was tracked by PC-based software that analyzed images from an overhead camera at 60 Hz. The time to first enter the electrified sector was assessed offline as a measure of spatial learning acquisition across the training trials. In addition, the number of shocks/entry was also tabulated as a measure of escape behavior to validate that there were no differences in pain threshold or sensorimotor behavior which would alter escape behavior across groups.

## MPORT

Animals were tested for learning and re-learning of spatial relationships using the hippocampal-dependent (*Barker and Warburton, 2011*) MPORT (multiple placement object recognition task, *Figure 5*) which assesses spatial memory based on the fact that mice naturally prefer novel object locations. This protocol is an variant of the human multiple placement task used to test re-learning in patients with neuropathologies, including those with autism (*D'Cruz et al., 2013*). Following an initial habituation to an empty arena for 1 hr and re-visit to the home cage (20 min), mice were allowed to examine 2 identical objects at opposite ends of the arena for 10 min (position 1). Following a 20 min re-visit to the home cage, mice were tested for two additional 10 min trials after one of the objects was re-located to two new positions (positions 2 and 3). All test trials were separated by a 20 min re-visit to the home cage.

The duration of examination (T) of the moved (M) and unmoved (U) objects were quantified. The discrimination ratio for detecting the moved object was quantified as: (T-M)/ (T-U). Both locomotor activity and total # approaches, a measure of interest in the objects, were also quantified across groups. Multiple trials were used where the location for one object was varied in order to test the ability of the animal to remember new locations. In experiments where spine type and number were quantified, animals were sacrificed by 1–2 hr after acclimation (naïve), learning trial 1 (learning) or learning trial 2 (re-learning). Behavioral data from animals used for spine typing was analyzed.

## Statistics

All data are presented as the mean ± the standard error of the mean (SEM) using Origin 8.5.1. A power analysis to determine the minimum sample size needed to achieve statistical significance was performed for all experiments achieving statistical significance (algorithms: http://www.originlab.com/doc/Origin-Help/PSS-ANOVA-Algorithm; http://www.originlab.com/doc/Origin-Help/PSS-t-Test2-Algorithm). Data were shown to fit a normal distribution using the Ko lmogorov-Smirnov test for normality, and Levene's test was used to confirm equal variance between groups being compared. All data were included in the analysis unless statistically defined as an outlier (>2 standard

deviations from the mean). Golgi, IHC, and behavioral experiments were performed in duplicate (exact n's are indicated in the figure legends). For Golgi and IHC experiments, 2–4 neurons were evaluated/mouse with 4–6 mice used per group. A statistically significant difference between groups for the LTD and LTP studies was determined by averaging EPSP slope in the final 20 min. for each recording; these numbers were averaged across groups and compared using the student's unpaired t-test. Comparisons of the degree of change across groups for all other experimental procedures were analyzed with a student's unpaired t-test (2 groups) or one-way analysis of variance (ANOVA, 3 + groups). Post-hoc comparisons for the ANOVA were made with a post-hoc Tukey's test. For all tests, the level of significance was determined to be $p < 0.05$. A complete description of the statistical analyses for all experiments (including n's, p values and power for significant findings) is detailed in the figure legends.

## Acknowledgements

We thank T Sacktor (SUNY Downstate) and P Bergold (SUNY Downstate) for a critical reading of the manuscript, G Homanics (Univ. Pittsburgh) for supplying the α4+/- mice and RE Mains (Univ. Conn. Health Center) for supplying the kalirin7 KO mice. We also thank A Mohammad and R George for helpful technical assistance. This work was supported by R01-MH100561 to SSS. The source data for all figures are included in the supplementary online material.

## Additional information

### Funding

| Funder | Grant reference number | Author |
| --- | --- | --- |
| National Institute of Mental Health | R01-MH100561 | Sheryl Sue Smith |

The funders had no role in study design, data collection and interpretation, or the decision to submit the work for publication.

### Author contributions

SA, Performed the spine density/morphological analyses and behavioral experiments, Analyzed the data and acquired images for +/+ and α4-/- and drug-treated mice, Contributed to the experimental design, Conception and design, Acquisition of data, Analysis and interpretation of data; JP, Performed the spine density/morphological analyses and behavioral experiments for the males and some of the drug groups, Performed the immunohistochemical experiments, Contributed to the experimental design, Conception and design, Acquisition of data, Analysis and interpretation of data; HS, Performed the electrophysiological experiments, Acquisition of data; SSS, Designed the experiments, Constructed the figures, Analysis and interpretation of data, Wrote the paper

### Author ORCIDs

Sheryl Sue Smith, http://orcid.org/0000-0003-4308-3267

### Ethics

Animal experimentation: This study was performed in strict accordance with the recommendations in the Guide for the Care and Use of Laboratory Animals of the National Institutes of Health. All of the animals were handled according to approved institutional animal care and use committee (IACUC) protocol (#13-10374) of SUNY Downstate Medical Center (Animal Welfare Assurance Number: A3260-01). All perfusions were performed under urethane anesthesia, and every effort was made to minimize suffering.

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
