## [Decision Letter]

Thank you for submitting your article "Pubertal synaptic pruning in hippocampus is triggered by α4βδ GABA-A receptor regulation of kalirin-7 spine remodeling" for consideration by *eLife*. Your article has been favorably evaluated by a Senior Editor and three reviewers, one of whom is a member of our Board of Reviewing Editors.

The reviewers have discussed the reviews with one another and the Reviewing Editor has drafted this decision to help you prepare a revised submission.

Summary:

Afroz and collaborators present a very interesting set of studies suggesting a fundamental role for α4 containing GABA_A_ receptors in adolescent synaptic pruning in CA1 from female mice. The most important new finding is that the previously observed expression of the GABA receptor, alpha4-β-δ on dendritic spines at the onset of puberty in female mice triggers pruning of synapses in area CA1 of the hippocampus that results in fewer mushroom shaped and stubby spines on CA1 pyramidal neurons in the post-pubertal period. Deletion of α4 containing GABA_A_ receptors prevents the pubertal pruning and results in spatial learning deficits after puberty. Furthermore, their results suggest a link between α4 inhibitory drive, resulting in decrease in NMDA receptor function, and decreased Kal7 expression in the initial steps leading to excitatory synaptic pruning in the female adolescent CA1.

Essential revisions:

For the most part, the reviewers found the data both sound and interesting. The work is expertly conducted and the results are generally unambiguous. However, all stated that the presentation is often inappropriate and needs major improvement. The presentation contains many instances of over-interpretation and inappropriate citation of the literature. In addition, the Introduction needs a clearer presentation of background information for the general audience of *eLife*.

1) The link to autism, which is mentioned four times in the Abstract, is not substantiated and is not necessary for the presentation. Indeed, it detracts from the interest of the data because it is not accurate. The work stands on its own as being important and does not need to be presented as relevant to a disease model. Because autism emerges in very young individuals (2-3 years old), a mechanism that occurs predominantly at puberty may be altered in some autistic patients, but it could not be causal. One paper cited by the authors suggests a link between synaptic pruning and autism, mediated by an mTor pathway, in the cortex. The other citation (Hoogenraad) does not mention synaptic pruning and should not be cited in this context. The link between autism and mutation of GABARalpha-4 is listed on the SFARI website as category 4: minimal evidence, linkage having been found in just two directed studies. There are over 100 gene mutations associated with ASD. Thus, GABARalpha4 isn't seen as having a strong linkage to autism. The link to autism should be removed from the Abstract and Introduction. It would be appropriate to mention it briefly as a possibility in the Discussion, along with a potential link to schizophrenia and other mental illnesses that emerge at adolescence.

2) The authors attribute the increase in spine density post-pubertally in the α4 KO to the increased expression of kalirin-7. This assertion is overstated and should be appropriately qualified. The authors state on page 10 that kalirin-7 is the only spine protein shown to alter spine density. This is not correct. As just one example, IQGAP1 has also been shown to regulate spine density (J. Neurosci. 31: 8533-8542, 2011). The authors need to improve their knowledge of the literature on regulation of spine density and modify this part of the text accordingly. They have shown that expression of kalirin-7 at puberty is changed in the α4 KO and it is appropriate to indicate that this change in expression may contribute to the change in spine density; however, it is not appropriate to attribute the entire effect to kalirin expression. The signaling pathways that control spine density are complex and various. To show that kalirin-7 is the critical link in this system, the authors would have to artificially express kalirin-7 in the CA1 pyramidal neurons at puberty in the WT mice and show that the exogenous over-expression eliminates the pruning. Alternatively, they would need to specifically knock-down kalirin-7 in the KO neurons and show that this leads to increased pruning. Since these experiments haven't been done, and would be difficult, the authors should simply indicate that the increased activation of NMDARs in the α4 KO enhances expression of kalirin-7 at spines, and *suggests* that kalirin-7 may be a critical part of the mechanism for initiating pruning during puberty.

3) The work has been performed in female mice, taking advantage of the increase in α4 in the easily recognized pubertal period. This is understandable, but it should be stated in the Abstract. Do the authors know whether the ensuing synaptic pruning is related exclusively to the female CA1? Is it known whether α4 containing GABA receptors emerge on dendrites during adolescence in males? Any specific link to females would be important to discuss in connection with the relationship of this pruning mechanism to neurodevelopmental disorders such as autism and schizophrenia. Autism is four times more prevalent in males than females. Similarly, a role for synaptic pruning was recently revisited in schizophrenia, as the authors cite. However, without knowing if the α4 GABAR-mediated pruning is active in males, it is difficult to make a connection to schizophrenia.

4) There are several clarifying issues to be addressed regarding the treatment with MK801.

A) Previous work on the effects of adolescent blockade of NMDA receptors by MK801 in rats has been shown to induce a reduced proportion of mushroom spines and an increased proportion of stubby spines in hippocampus (Han et al., 2013 Psychiatry Res.210(1):351-6). This work is not cited.

B) The length of MK801 treatment is not described. Was it one injection? Or, one daily during the adolescent period? In general: What is the time course of the drug administration in terms of remodeling synapses, i.e. how long does pruning take? This was not immediately self-evident.

C) MK801 treatment in vivo during adolescence affects excitatory neurons, but also affects the inhibitory system, producing alterations that are different in each cell type (Thomases et al., 2013. J Neurosci. 33(1):26-34; Wang and Gao 2012. Neuropharmacology 62(4):1808-22). Systemic administration of NMDA receptor antagonists would be expected to produce global alterations in the excitatory/inhibitory balance, which are not solely related to increased NMDA receptor subunits in dendritic spines.

D) Figure 2—figure supplement 1 is not a valid measure of increased expression of NR1 after treatment with MK-801. Detection on western blots by chemiluminescence is extremely non-linear and thus not reliably quantitative. The signals are easily saturated; thus, the GAPDH signal could be saturated and not a valid loading control.

The authors should eliminate Figure 2—figure supplement 1, and simply mention a change in NR1 expression as a possibility. They should edit and temper the discussion of this experiment as noted in A, B and C.

5) The choice of active avoidance and MPORT to test for behavioral flexibility is not justified. Given that these tests have a strong spatial component, and what is being studied is the hippocampus, it is understandable that these behavioral tasks were chosen. However, behavioral flexibility is the result of the interaction of many brain regions, not just the hippocampus. The difference in learning capacity between wt and KO is evident, but may not be a reflection of overall behavioral flexibility. The presentation of these results should be modified. In addition, the authors present evidence of synaptic remodeling 2 hours after learning, which is on par in magnitude with that seen following pruning at adolescence. This highlights how dramatically plastic the spines are and raises the question of the significance of the pruning at puberty if the same parameter can also be so quickly modified by a simple learning task. How much of the current results are due to the depauperate conditions under which mice are housed? Under normal conditions animals would be learning all the time. Can the authors discuss this issue?

6) In general, the Introduction to the work is not adequate. The Introduction should concisely summarize the prior work showing that alpha4-β-δ receptors emerge on dendrites at puberty, and the previous studies, largely from the authors' lab, showing that the presence of the receptors impairs spatial learning during the pubertal period. They should say that the authors have now extended that work to show that the presence of the receptors initiates synaptic pruning which results in lower spine density in CA1 in the post-pubertal period. Puberty, and the post-pubertal period should be defined in the Introduction; it shouldn't wait for the figure legends. One reviewer had to read the paper twice to sort out the story. Some of the senior author's review articles (e.g. Br. Res., 2016) should be cited in the Introduction for context. In the Introduction, the authors state, "An optimal spine density, produced by developmental pruning of unnecessary synapses, may be necessary not only for the ability to form memories but also the ability to re-learn or "update" previously learned information," and cite a modeling study. This role of spine density has not been shown experimentally. The sentence should clearly say that "A modeling study suggests…". Again, in the last paragraph of the Discussion: "Removal of unnecessary spines during adolescence is thought to be necessary for optimal cognitive ability (Meilijson and Ruppin, 1999)”. This reference is a modeling study, with many unsupported assumptions, and is not an adequate reference to establish this statement. Is there an appropriate experimental study?

---

## [Author Response]

Essential revisions:

For the most part, the reviewers found the data both sound and interesting. The work is expertly conducted and the results are generally unambiguous. However, all stated that the presentation is often inappropriate and needs major improvement. The presentation contains many instances of over-interpretation and inappropriate citation of the literature. In addition, the Introduction needs a clearer presentation of background information for the general audience of eLife.

1) The link to autism, which is mentioned four times in the Abstract, is not substantiated and is not necessary for the presentation. Indeed, it detracts from the interest of the data because it is not accurate. The work stands on its own as being important and does not need to be presented as relevant to a disease model. Because autism emerges in very young individuals (2-3 years old), a mechanism that occurs predominantly at puberty may be altered in some autistic patients, but it could not be causal. One paper cited by the authors suggests a link between synaptic pruning and autism, mediated by an mTor pathway, in the cortex. The other citation (Hoogenraad) does not mention synaptic pruning and should not be cited in this context. The link between autism and mutation of GABARalpha-4 is listed on the SFARI website as category 4: minimal evidence, linkage having been found in just two directed studies. There are over 100 gene mutations associated with ASD. Thus, GABARalpha4 isn't seen as having a strong linkage to autism. The link to autism should be removed from the Abstract and Introduction. It would be appropriate to mention it briefly as a possibility in the Discussion, along with a potential link to schizophrenia and other mental illnesses that emerge at adolescence.

We have removed all references to autism from the Abstract and Introduction, as suggested. Instead, the possible relevance of our findings to autism and schizophrenia is discussed in the final paragraph of the paper.

2) The authors attribute the increase in spine density post-pubertally in the α4 KO to the increased expression of kalirin-7. This assertion is overstated and should be appropriately qualified. The authors state on page 10 that kalirin-7 is the only spine protein shown to alter spine density. This is not correct. As just one example, IQGAP1 has also been shown to regulate spine density (J. Neurosci. 31: 8533-8542, 2011). The authors need to improve their knowledge of the literature on regulation of spine density and modify this part of the text accordingly. They have shown that expression of kalirin-7 at puberty is changed in the α4 KO and it is appropriate to indicate that this change in expression may contribute to the change in spine density; however, it is not appropriate to attribute the entire effect to kalirin expression. The signaling pathways that control spine density are complex and various. To show that kalirin-7 is the critical link in this system, the authors would have to artificially express kalirin-7 in the CA1 pyramidal neurons at puberty in the WT mice and show that the exogenous over-expression eliminates the pruning. Alternatively, they would need to specifically knock-down kalirin-7 in the KO neurons and show that this leads to increased pruning. Since these experiments haven't been done, and would be difficult, the authors should simply indicate that the increased activation of NMDARs in the α4 KO enhances expression of kalirin-7 at spines, and suggests that kalirin-7 may be a critical part of the mechanism for initiating pruning during puberty.

We have re-written the paragraph of the Discussion which describes the kalirin-7 findings to include many of the other spine proteins. We also have changed the wording to “suggest” rather than state that kalirin may be a mechanism for our findings.

3) The work has been performed in female mice, taking advantage of the increase in α4 in the easily recognized pubertal period. This is understandable, but it should be stated in the Abstract. Do the authors know whether the ensuing synaptic pruning is related exclusively to the female CA1? Is it known whether α4 containing GABA receptors emerge on dendrites during adolescence in males? Any specific link to females would be important to discuss in connection with the relationship of this pruning mechanism to neurodevelopmental disorders such as autism and schizophrenia. Autism is four times more prevalent in males than females. Similarly, a role for synaptic pruning was recently revisited in schizophrenia, as the authors cite. However, without knowing if the α4 GABAR-mediated pruning is active in males, it is difficult to make a connection to schizophrenia.

We have performed the experiments in males and now include synaptic pruning data on males in the supplementary material (both wild-type and α4-/-). We plan a separate detailed paper on the male data, showing increased expression of α4βδ GABA_A_ receptors, and thus only spine density measurements are included in this manuscript. We also discuss the male data with respect to potential relevance of our data to autism and schizophrenia. Because most of the data is focused on the female mouse, we now state this in the Abstract and title.

4) There are several clarifying issues to be addressed regarding the treatment with MK801.

A) Previous work on the effects of adolescent blockade of NMDA receptors by MK801 in rats has been shown to induce a reduced proportion of mushroom spines and an increased proportion of stubby spines in hippocampus (Han et al., 2013 Psychiatry Res.210(1):351-6). This work is not cited.

We now cite the work of Han et al., 2013, describing effects on MK-801 to decrease mushroom spines but increase stubby spines. Although the results are not identical to ours, there are potential age and gender differences which may explain the differences.

B) The length of MK801 treatment is not described. Was it one injection? Or, one daily during the adolescent period? In general: What is the time course of the drug administration in terms of remodeling synapses, i.e. how long does pruning take? This was not immediately self-evident.

MK-801 was administered once a day throughout the 10 d pubertal period (PND 35-44). This was the protocol for all of the drug protocols to alter synaptic pruning. This protocol is now more clearly stated, both in the Methods, as well as in the Results.

C) MK801 treatment in vivo during adolescence affects excitatory neurons, but also affects the inhibitory system, producing alterations that are different in each cell type (Thomases et al., 2013. J Neurosci. 33(1):26-34; Wang and Gao 2012. Neuropharmacology 62(4):1808-22). Systemic administration of NMDA receptor antagonists would be expected to produce global alterations in the excitatory/inhibitory balance, which are not solely related to increased NMDA receptor subunits in dendritic spines.

We have now added the additional suggested references showing that MK-801 also affects the GABAergic system, and thus has a more complex effect on the excitatory/inhibitory balance, which may have acted to alter spine density.

D) Figure 2 —figure supplement 1 is not a valid measure of increased expression of NR1 after treatment with MK-801. Detection on western blots by chemiluminescence is extremely non-linear and thus not reliably quantitative. The signals are easily saturated; thus, the GAPDH signal could be saturated and not a valid loading control.

The authors should eliminate Figure 2—figure supplement 1, and simply mention a change in NR1 expression as a possibility. They should edit and temper the discussion of this experiment as noted in A, B and C.

Figure 2—figure supplement 1 has been deleted. The change in receptors is now mentioned as a possible reason (with supporting citations), and the discussion now includes many other possibilities for the actions of MK-801 (as in A-C).

5) The choice of active avoidance and MPORT to test for behavioral flexibility is not justified. Given that these tests have a strong spatial component, and what is being studied is the hippocampus, it is understandable that these behavioral tasks were chosen. However, behavioral flexibility is the result of the interaction of many brain regions, not just the hippocampus. The difference in learning capacity between wt and KO is evident, but may not be a reflection of overall behavioral flexibility. The presentation of these results should be modified. In addition, the authors present evidence of synaptic remodeling 2 hours after learning, which is on par in magnitude with that seen following pruning at adolescence. This highlights how dramatically plastic the spines are and raises the question of the significance of the pruning at puberty if the same parameter can also be so quickly modified by a simple learning task. How much of the current results are due to the depauperate conditions under which mice are housed? Under normal conditions animals would be learning all the time. Can the authors discuss this issue?

As suggested, we have modified our discussion of the behavioral findings as reflecting learning and re-learning rather than behavioral flexibility. We also discuss the plasticity of spines with regards to learning changes versus pubertal changes – both are of similar magnitude, although they produce different changes in spine types (decreases in mushroom spines at puberty versus increases in mushroom spines after learning). Although this does indeed suggest that the spines are remarkably plastic, pubertal pruning is notable because it alone results in reduced numbers of the normally stable mushroom spines. We have added this to the Discussion. As suggested, we also include a discussion of the fact that animal cages were not enriched, and therefore the spine density of control animals likely reflects the lower limit because enrichment would be expected to increase spine density.

6) In general, the Introduction to the work is not adequate. The Introduction should concisely summarize the prior work showing that alpha4-β-δ receptors emerge on dendrites at puberty, and the previous studies, largely from the authors' lab, showing that the presence of the receptors impairs spatial learning during the pubertal period. They should say that the authors have now extended that work to show that the presence of the receptors initiates synaptic pruning which results in lower spine density in CA1 in the post-pubertal period. Puberty, and the post-pubertal period should be defined in the Introduction; it shouldn't wait for the figure legends. One reviewer had to read the paper twice to sort out the story. Some of the senior author's review articles (e.g. Br. Res., 2016) should be cited in the Introduction for context. In the Introduction, the authors state, "An optimal spine density, produced by developmental pruning of unnecessary synapses, may be necessary not only for the ability to form memories but also the ability to re-learn or "update" previously learned information," and cite a modeling study. This role of spine density has not been shown experimentally. The sentence should clearly say that "A modeling study suggests…". Again, in the last paragraph of the Discussion: "Removal of unnecessary spines during adolescence is thought to be necessary for optimal cognitive ability (Meilijson and Ruppin, 1999)”. This reference is a modeling study, with many unsupported assumptions, and is not an adequate reference to establish this statement. Is there an appropriate experimental study?

We have re-written the Introduction to include a more detailed description of the previous studies from our lab (Shen et al., 2007; Shen et al., 2010; Shen et al., 2016) and how the present study is an extension of this work. We also define puberty (PND 35-44) and the post-pubertal (PND 56) periods in the Introduction.

We also have re-written the line in the Introduction indicating that a modelling study suggests that a high spine density impairs re-learning. Although many studies report that spine loss is correlated with learning deficits, we were unable to find experimental studies reporting that increases in spine density are correlated with learning deficits. We now point out in the Discussion that our findings, showing learning deficits in animals with greater spine density, may be one of the first experimental demonstrations of what has only been suggested in modelling studies.